# Low input capture Hi-C (liCHi-C) identifies promoter-enhancer interactions at high-resolution

Laureano Tomás-Daza[1,2,14], Llorenç Rovirosa[1,14], Paula López-Martí[1,2], Andrea Nieto-Aliseda[1], François Serra [1], Ainoa Planas-Riverola[1], Oscar Molina [1], Rebecca McDonald [3], Cedric Ghevaert[3,4], Esther Cuatrecasas[5], Dolors Costa[6,7,8], Mireia Camós[9,10,11], Clara Bueno[1], Pablo Menéndez[1,12], Alfonso Valencia [2,12] & Biola M. Javierre [1,13] ✉

Long-range interactions between regulatory elements and promoters are key in gene transcriptional control; however, their study requires large amounts of starting material, which is not compatible with clinical scenarios nor the study of rare cell populations. Here we introduce low input capture Hi-C (liCHi-C) as a cost-effective, flexible method to map and robustly compare promoter interactomes at high resolution. As proof of its broad applicability, we implement liCHi-C to study normal and malignant human hematopoietic hierarchy in clinical samples. We demonstrate that the dynamic promoter architecture identifies developmental trajectories and orchestrates transcriptional transitions during cell-state commitment. Moreover, liCHi-C enables the identification of disease-relevant cell types, genes and pathways potentially deregulated by non-coding alterations at distal regulatory elements. Finally, we show that liCHi-C can be harnessed to uncover genome-wide structural variants, resolve their breakpoints and infer their pathogenic effects. Collectively, our optimized liCHi-C method expands the study of 3D chromatin organization to unique, low-abundance cell populations, and offers an opportunity to uncover factors and regulatory networks involved in disease pathogenesis.

Enhancers are critical modulators of gene transcription through physical interactions with target promoters that often locate distally in the genome. The physical proximity between enhancers and promoters is ultimately enabled and determined by the three-dimensional folding of the chromatin within the nucleus[1,2]. Although enhancers can be defined through well-characterized features, predicting their target genes at distal locations remains challenging due to the high complexity of studying enhancer–promoter interactions and the large

[1]Josep Carreras Leukaemia Research Institute, Badalona, Barcelona, Spain. [2]Barcelona Supercomputing Center, Barcelona, Barcelona, Spain. [3]Wellcome-MRC Cambridge Stem Cell Institute, Cambridge, UK. [4]NHS Blood and Transplant, Cambridge, UK. [5]Pediatric Institute of Rare Diseases, Sant Joan de Déu Hospital, Esplugues de Llobregat, Barcelona, Spain. [6]Hospital Clinic, Barcelona, Spain. [7]Institute of Biomedical Research August Pi i Sunyer, Barcelona, Spain. [8]Cancer Network Biomedical Research Center, Barcelona, Spain. [9]Sant Joan de Déu Research Institute, Esplugues de Llobregat, Barcelona, Spain. [10]Sant Joan de Déu Hospital, Esplugues de Llobregat, Barcelona, Spain. [11]Center for Biomedical Research in the Rare Diseases Network (CIBERER), Carlos III Health Institute, Madrid, Spain. [12]Catalan Institution for Research and Advanced Studies (ICREA), Barcelona, Spain. [13]Institute for Health Science Research Germans Trias i Pujol, Badalona, Barcelona, Spain. [14]These authors contributed equally: Laureano Tomás-Daza, Llorenç Rovirosa. ✉e-mail: bmjavierre@carrerasresearch.org

variability according to cell type and state. This gap of knowledge is particularly problematic for understanding the molecular mechanisms associated with inherited and de novo acquired mutations and epi-mutations involved in common human diseases, which are all highly enriched at regulatory elements[3,4].

To enable the study of genomic regulatory mechanisms underlying disease pathologies at a genome-wide scale, we previously developed the promoter capture Hi-C (PCHi-C) method[5,6]. This approach allows systematic identification of the promoter interactome (i.e., genomic regions, including distal regulatory regions, in physical proximity with more than 31,000 promoters) independently of the activity status of the interacting regions. This method has allowed us to uncover aspects of the diversity of transcriptional regulatory factors[7] and mechanisms in cell differentiation[8,9] and disease[5], and it has broadened our capacity to identify hundreds of potential disease-candidate genes and/or gene pathways potentially deregulated by noncoding disease-associated variants[5,10–21]. However, PCHi-C relies on the availability of millions of cells, typically ranging between 30 and 50 million cells per biological replicate, which prohibits the analysis of rare cell populations such as those commonly obtained in clinical settings.

Here, to overcome this limitation, we present liCHi-C, a mini-input method that allows the generation of high-resolution genome-wide promoter interactome maps using very low amounts of starting material. We have validated our method by benchmarking liCHi-C promoter interactomes against the highest resolution PCHi-C promoter interactomes available to date, demonstrating that the interactomes can be reproducibly interrogated using as low as 50,000 cells of starting material. As a proof of its potential for discovering insights about gene transcription regulation, we used liCHi-C to study human hematopoiesis in vivo and demonstrate its potential for identifying developmental trajectories and providing mechanistic understanding of transcriptional dynamics along in vivo cell commitment. Furthermore, we show that liCHi-C can be applied to investigate molecular links between disease-associated noncoding alterations at distal regulatory elements with their target genes in rare cell populations that cannot be characterized using PCHi-C. Finally, to support the broad applicability of liCHi-C across disease settings, we analyze primary leukemias and identify patient-specific structural genomic alterations and cancer-specific topological features potentially implicated in gene deregulation and disease etiology. All the computational tools to analyze and integrate liCHi-C data are freely available at https://github.com/JavierreLab/liCHiC.

## Results

### Development and optimization of liCHi-C for low-input samples
In order to enable the detection of the promoter interactome using low-input material, we modified the original PCHi-C to minimize losses during the procedure. Specifically, liCHi-C maximizes library complexity by employing a single tube, modifying reagent concentration and volume, and eliminating or modifying the sequence of some steps (Fig. 1A). In addition, it reduces by half the time spent on the library preparation. For more details, see "Methods" and Supplementary Fig. 1A.

To systematically evaluate liCHi-C, we generated libraries from decreasing numbers of human naive B cells (Supplementary Data 1) at controlled ratios and compared these with the most comprehensive PCHi-C data available to date from the exact same cell type that used ~40 million (40 M) cells as starting material. Each sample was deep-sequenced and paired-end reads were mapped and filtered using HiCUP pipeline[22]. Visual inspection of normalized liCHi-C and PCHi-C contact maps showed a high degree of similarity in topological properties (Supplementary Fig. 1B), and for all experimental conditions (i.e., number of starting cells), the percentage of valid reads (mean = 58.41%; SD = 5.76%) and the capture efficiency (mean = 61.29%; SD = 9.15%) were similar between both methods, being 23.58% (SD = 1.59%) of

those contacts in *trans* (Supplementary Fig. 1C–E and Supplementary Data 1). Unsurprisingly, given the higher need for amplification, the PCR duplicates increased with lower amounts of starting cells, ranging from 86% in 50k cell samples, to 12% of the mapped reads in 40 M cell samples. Interaction matrices were also highly reproducible for both methods and different amounts of starting material (stratum-adjusted correlation coefficient (SCC) >0.90 at 100 kb resolution) (Fig. 1B and Supplementary Fig. 1F). These data suggest that liCHi-C reliably generates high-quality promoter interactomes, including 31,253 annotated promoters, and can routinely be performed successfully with an input as low as 50k cells. Remarkably, liCHi-C was able to achieve a >tenfold enrichment of read pairs involving promoters when compared with Hi-C using 800 times less of the starting cells (Supplementary Data 1).

### Comparison of liCHi-C with other C-based methods for profiling promoter interactomes
To formally compare the performance of liCHi-C to detect promoter interactions, we used the CHiCAGO pipeline[23] to call for significant interactions (CHiCAGO score >5). Distance distribution and nature of interacting fragments were similar across cell number conditions and methods (Supplementary Fig. 2A–C). Specifically, we found a median linear distance between promoters and their interacting regions of 265 kb (SD = 30 kb), and 89.04% (SD = 3.72%) of these were promoter-to-non-promoter interactions (Supplementary Fig. 2B, C and Supplementary Data 1). Principal component analysis (PCA) of CHiCAGO interaction scores across all biological replicates demonstrated that patterns of promoter interactions are highly consistent across biological replicates and group samples according to the number of input cells (Fig. 1C). To further explore the limits of liCHi-C library complexity we performed hierarchical clustering based on their CHiCAGO interaction scores. We observed that promoter interactomes generated from >100k cells reproduce the ones generated with PCHi-C on 40M cells (Supplementary Fig. 2D). Although promoter interactomes with less than 100k cells showed a different clustering profile, potentially reflecting the reduction of significant interactions due the limited library complexity, these retain cell-type specific and invariant topological features (Fig. 1D and Supplementary Fig 2E–G). Collectively, these results demonstrate the high reproducibility of liCHi-C to profile promoter interactomes at a similar resolution as PCHi-C, and underscore the suitability of the approach to investigate these interactions in cell populations present at relatively low abundance within a sample.

In addition, we benchmarked liCHi-C against other existing C-based methods used for detecting the 3D genome topologies of blood lineage, including Low-C[24], Hi-C (using a four-cutter[25] or a six-cutter[5] restriction enzyme) and TagHi-C[26] (Supplementary Fig 3 and Supplementary Data 2). We called chromatin loops in these datasets at a resolution of 5 kb, which is similar to the resolution of our liCHi-C data (~4096 bp), using HICCUPS[27], Mustache[28], and HiCExplorer[29] loop callers with standard parameters (for more details, see "Methods"). Despite the fact that these methods have provided fundamental insights about the role and regulation of spatio-temporal genome architecture on rare cell populations, these methods are very limited with respect to the detection of promoter interactions at restriction fragment resolution (Supplementary Fig. 3A, B and Supplementary Data 2). Collectively, these results demonstrate that liCHi-C outperforms other existing C-based methods at genome-wide and promoter-wide detection of potential gene-regulatory interactions at high resolution using low cell numbers.

### liCHi-C efficiently captures promoter interactomes across different hematopoietic lineages
To assess the capacity of liCHi-C to provide fundamental insight about in vivo cell differentiation, we performed high-quality liCHi-C experiments in nine distinct cellular populations from the human hematopoietic hierarchy (two biological replicates per cell type; 500k cells per

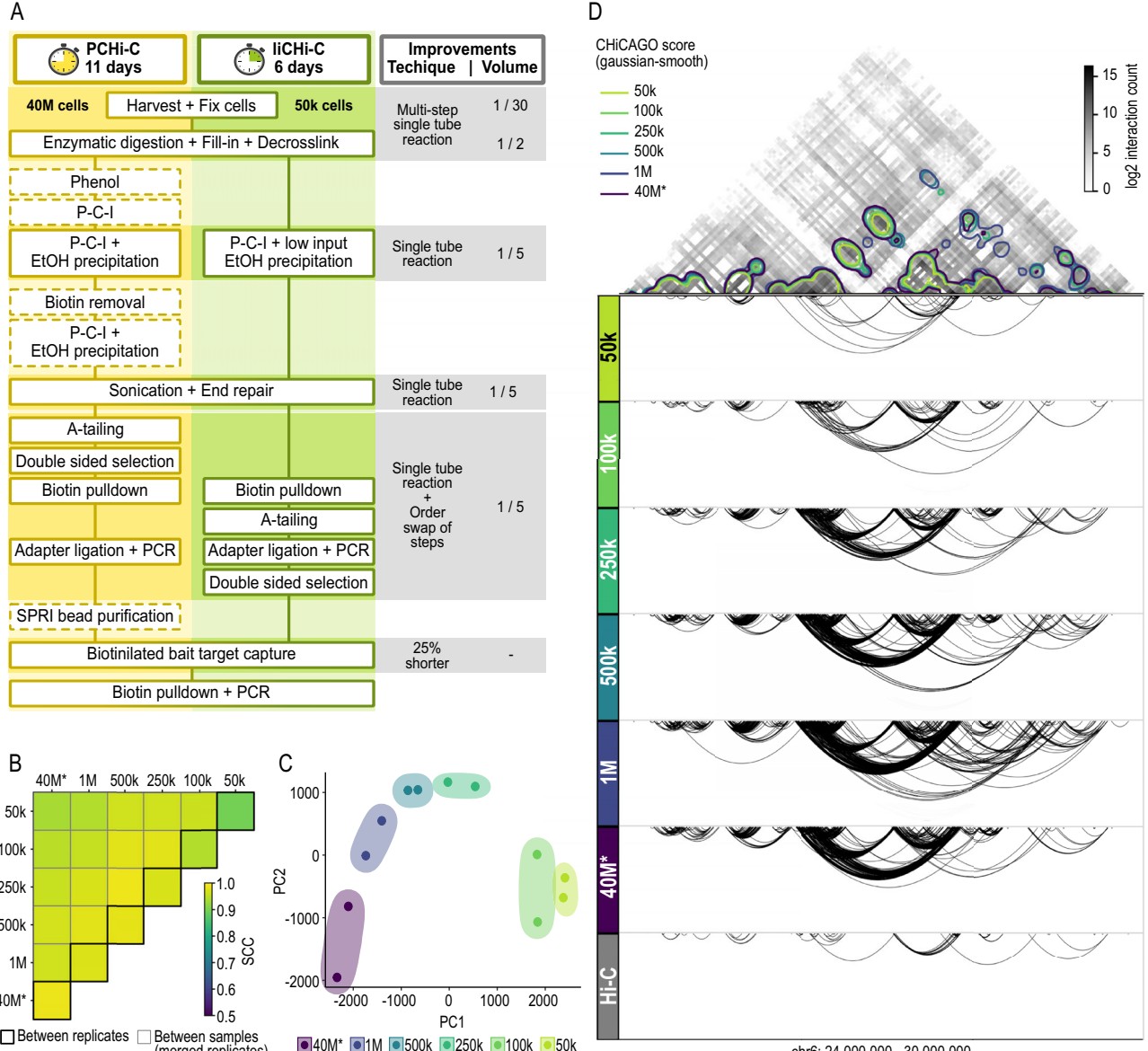

**Fig. 1 | liCHi-C is a robust and reproducible method to study the promoter interactome in low-abundance cell populations. A** Schematic comparison of the PCHi-C vs liCHi-C workflow. Dash-bordered boxes denote removed steps from the original PCHi-C protocol. **B** Heatmap displaying stratum-adjusted correlation coefficient (SCC) between promoter interactomes of human naive B cells obtained by liCHi-C and PCHi-C (*) using different cell numbers at 100 kb resolution. Reproducibility between biological replicates and between pairs of merged biological replicates are surrounded by black and gray lines, respectively. k 1000, M million. **C** Principal Component Analysis of CHiCAGO significant interactions (CHiCAGO scores >5) of biological replicates detected by liCHi-C and PCHi-C (*) using different cell numbers. **D** Top: Interaction matrix at 50 kb resolution generated with PCHi-C and 40 million cells. Colored contour plot over the interaction matrix represents gaussian smoothing (alpha = 1.2) of the significant CHiCAGO interactions detected by liCHi-C data using different numbers of input cells. Bottom: Significant interactions (arcs) detected with liCHi-C and PCHi-C (*) and Hi-C using different number of input cells.

replicate), including hematopoietic stem and progenitor cells (HSC), common myeloid progenitors (CMP), common lymphoid B-cell progenitors (CLP) and six differentiated cell types (Supplementary Fig. 4A–D and Supplementary Data 1). As a more comprehensive validation of the liCHi-C method, we first focused on the differentiated cell types for which high quality PCHi-C data is available. Benchmarking of liCHi-C data against PCHi-C data demonstrated high reproducibility between both methods for all profiled cell types (SCC > 0.93) (Fig. 2A and Supplementary Fig. 4D). Moreover, promoter interactomes clearly separated cell types independently of the method used (Supplementary Fig. 4G).

We then analyzed liCHi-C data in detail. Applying CHiCAGO we identified a median of 134,965 high-confidence promoter interactions (CHiCAGO score >5) per cell type (Supplementary Fig. 4E, F and

Supplementary Data 1). A PCA of interaction scores demonstrated that promoter interactomes were highly reproducible and cell-type specific (Fig. 2B). While the first principal component separated cells according to their myeloid or lymphoid linage, the second principal component recapitulated the differentiation potential, allowing altogether to identify both myeloid and lymphoid differentiation trajectories. To decipher these specificities in greater depth we applied AutoClass Bayesian clustering and computed the specificity score of each cluster in each cell type (Fig. 2C, D). Among the 33 clusters, we observed a stem and progenitor-specific cluster (C3), lymphoid-specific clusters (e.g., C10, C13), myeloid-specific clusters (e.g., C25, C27) and cell-type specific clusters (e.g., C4, C7) that contained genes known to be involved in cellular functions important for the given cell types (Fig. 2D–G). These observations were clearly illustrated by the

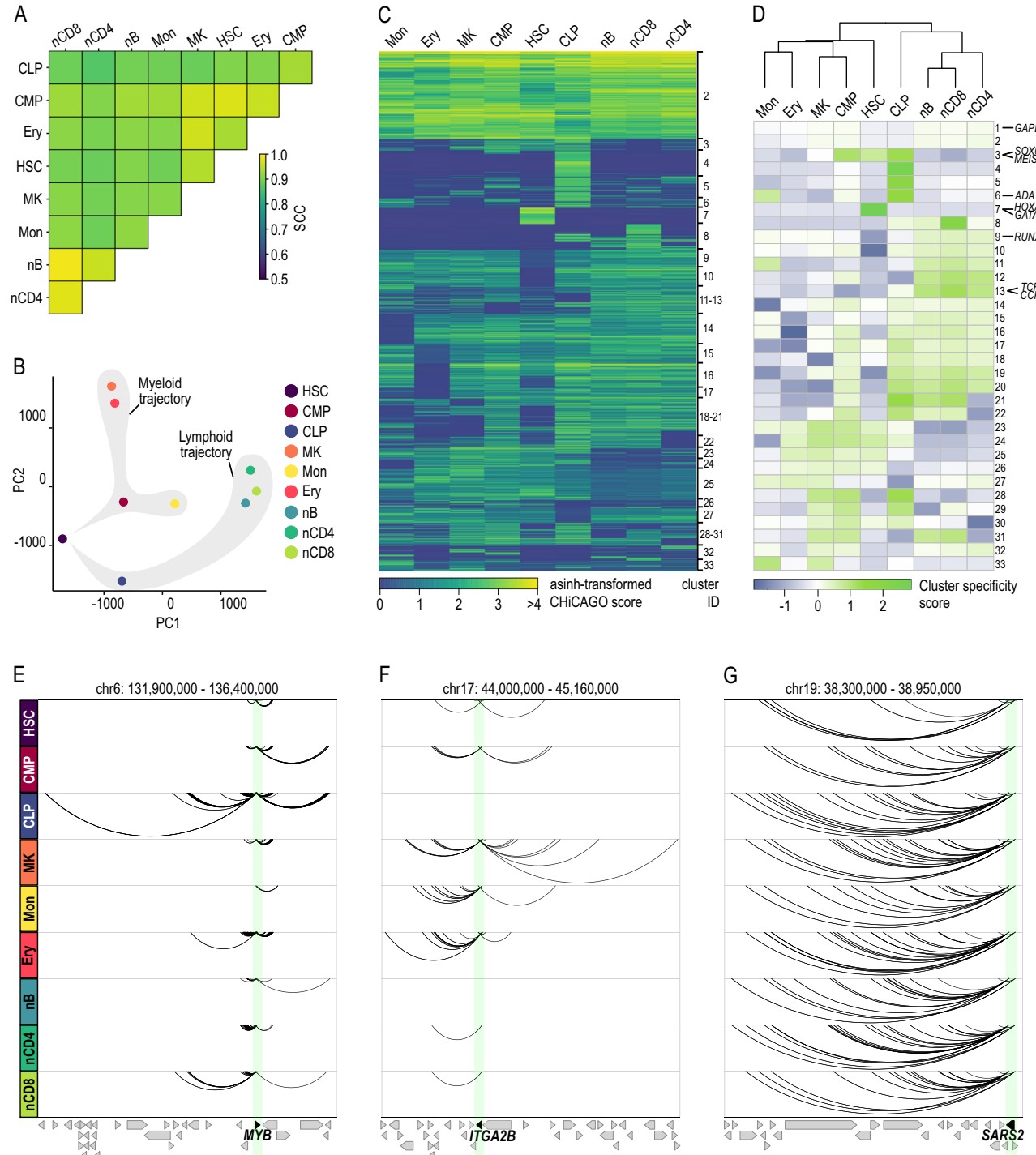

**Fig. 2 | liCHi-C enables the study of in vivo human hematopoiesis. A** Heatmap displaying stratum-adjusted correlation coefficient (SCC) between promoter interactomes of merged biological replicates. HSC hematopoietic stem cell, CMP common myeloid progenitor, CLP common B-cell lymphoid progenitor, MK megakaryocytes, Mon monocytes, Ery erythroblast, nB naive B cell, nCD4 naive CD4[+] cells, nCD8 naive CD8[+] cells. **B** Principal component analysis of liCHi-C significant interactions (CHiCAGO scores >5) from merged biological replicates. Shaded in gray are the predicted differentiation trajectories for both lymphoid and myeloid lineages. **C** Heatmap of asinh-transformed CHiCAGO score of significant interactions in at least one cell-type clustered using Autoclass algorithm. **D** Top: Dendrogram of hierarchical clustering with average linkage based on Euclidean distances of CHiCAGO significant interactions of merged samples. Bottom: Heatmap of cluster specificity score of each Autoclass cluster. *MYB* (**E**), *ITGA2B* (**F**), and *SARS2* (**G**) promoter-centered interactions (arcs) according to liCHi-C data. Arrows symbolize gene placement and orientation along the genomic window. Green shade depicts the gene promoter.

promoter interactomes of *MYB*[30], which encodes for a transcription regulator that plays an essential role in the regulation of lymphoid priming and early B-cell development (Fig. 2E), *ITGA2B*[31], which encodes for the megakaryocyte-specific surface marker CD41 (Fig. 2F), and *SARS2*[32], a housekeeping gene that encodes for the mitochondrial seryl-tRNA synthetase (Fig. 2G). Collectively, this data demonstrates that promoter interactomes are specific to the differentiation trajectories of cell types, and further suggests that the highly dynamic promoter-centric genome architecture recapitulates the developmental history of hematopoietic cell lineages.

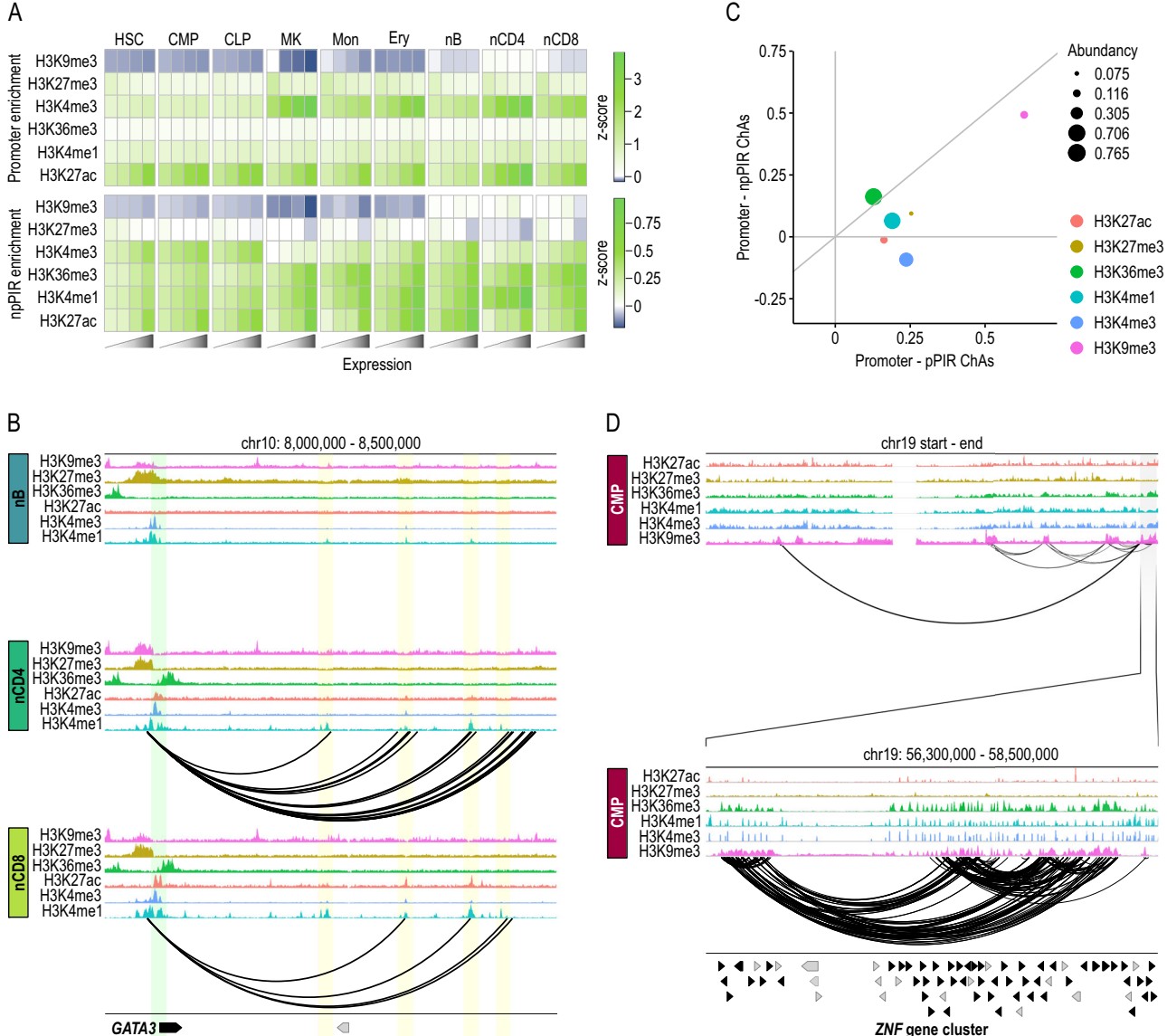

**Fig. 3 | liCHi-C identifies enhancer–promoter pairs and proteins involved in chromatin organization. A** Heatmap representing enrichment (expressed in terms of z-scores) of promoters (top) and non-promoter promoter-interacting regions (npPIRs) (bottom) for histone marks. PIRs and promoters were classified by the quartiles level of FPKM expression of the associated gene. HSC hematopoietic stem cell, CMP common myeloid progenitor, CLP common B-cell lymphoid progenitor, MK megakaryocytes, Mon monocytes, Ery erythroblast, nB naive B cell, nCD4 naive CD4+ cells, nCD8 naive CD8+ cells. **B** *GATA3* regulatory landscape in naive B (top), naive CD4+ T (middle), and naive CD8+ T (bottom) cells according to histone modifications and liCHi-C data. Green shade depicts the gene promoter, while yellow shades depict putative enhancer regions for that gene in any of these cell types. Arrows symbolize gene placement and orientation along the genomic window. **C** Scatterplot plot of chromatin assortativity (ChAs) of six histone marks computed for the promoter−pPIR sub-network (*x* axis) against the promoter−npPIR (*y* axis) for common myeloid progenitor cells. Dot size depends on the abundancy of each histone mark. **D** H3K9me3-centered interaction network (arcs) for common myeloid progenitor cells. Top graph depicts whole chromosome 19, while bottom graph shows a zoom centered on a *ZNF* gene cluster. Arrows symbolize gene placement and orientation along the genomic window.

## Promoter interactomes reshape transcriptional trajectories during in vivo cell commitment

To validate the ability of liCHi-C to uncover mechanistic insights on transcription regulation, we computationally integrated promoter interactome data with RNA-seq and ChIP-seq data from matched cell types (Supplementary Fig. 5A–C and Supplementary Data 3). We distinguish between two types of promoter-interacting regions (PIRs): non-promoter PIRs (npPIRs), in which the promoter-interacting region does not contain any captured gene promoter, and promoter PIRs (pPIRs), in which the promoter interacting region contains at least a gene promoter. We found high enrichment of histone modifications indicative of active enhancers (e.g., H3K27ac, H3K4me1) and noncoding transcription of regulatory regions (e.g., H3K4me3)[33] at distal npPIRs. These enrichments were positively associated with the expression level of linked genes in a cell-type-specific manner (Fig. 3A). Conversely, npPIRs of lowly expressed genes tend to be more enriched in repressive histone marks, such as H3K27me3 and H3K9me3, than more highly expressed ones. These enrichment profiles were highly consistent with the ones obtained with PCHi-C data despite the significant difference in the starting cell number (Supplementary Fig. 5D). Collectively, these results, exemplified by the transcriptional regulation of the T-cell-specific gene *GATA3*[34] (Fig. 3B), the B-cell-specific gene *PAX5*[35] (Supplementary Fig. 5E), and the HSC-specific gene *CD34*[36] (Supplementary Fig. 5F), demonstrate the capacity of liCHi-C to identify distal regulatory elements for each gene in rare cell types, and suggest that promoter-associated regions are enriched in distal

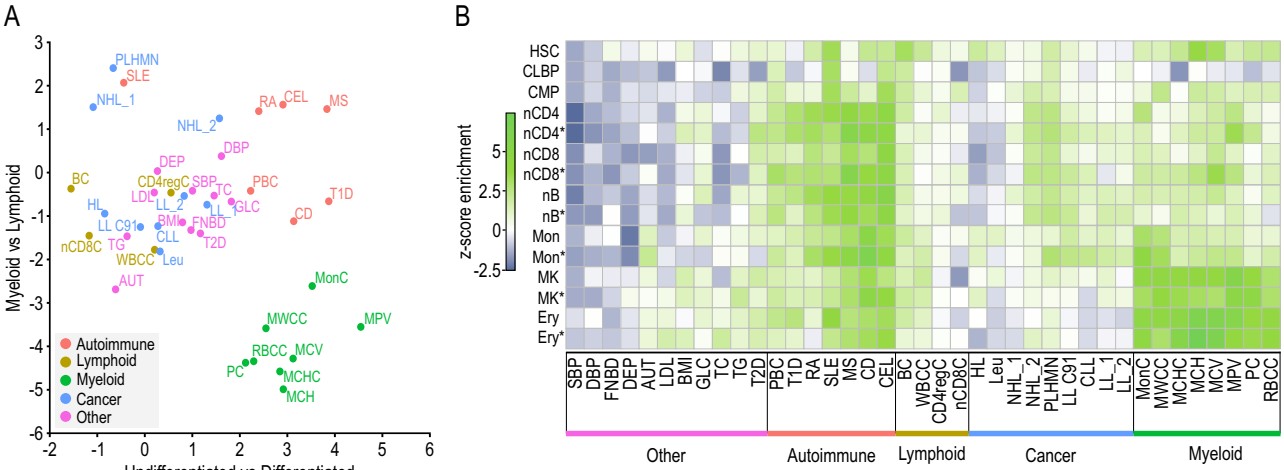

**Fig. 4 | liCHi-C identifies cell types presumably implicated in disease etiology.** **A** Enrichment of GWAS summary statistics at promoter-interacting regions (PIRs) by cell type. Axes reflect Blockshifter z-scores for two different tissue group comparisons: myeloid versus lymphoid (*y* axis) and undifferentiated against differentiated cells (*x* axis). Traits are labeled and colored by category: autoimmune disorder (red), lymphoid trait (yellow), myeloid trait (green), blood cancer (blue), and other (purple). SBP systolic blood pressure, DBP diastolic blood pressure, FNBD femoral neck bone mineral density, DEP depression, AUT autism, LDL low-density lipoprotein, BMI body mass index, GLC glucose sensitivity, TC total cholesterol, TG triglycerides, T2D type II diabetes, PBC primary biliary cirrhosis, T1D type I diabetes, RA rheumatoid arthritis, SLE systemic lupus erythematosus, MS multiple sclerosis, CD Crohn's disease, CEL celiac disease, BC B-cell absolute count, WBCC white blood cell count, CD4reGC CD4 regulatory T-cell absolute count, nCD8C naive CD8+ T-cell absolute count, HL Hodgkins lymphoma, Leu leukemia, NHL_1 non-Hodgkins lymphoma_1, NHL_2 non-Hodgkins lymphoma_2, PLHMN primary lymphoid and hematopoietic malignant neoplasms, LLC91 lymphoid leukemia, CLL chronic lymphocytic leukemia, LL_1 lymphoid leukaemia_1, LL_2 lymphoid leukaemia_2, MC monocyte count, MWCC myeloid white cell count, MCHC mean corpuscular hemoglobin concentration, MCH mean corpuscular hemoglobin, MCV mean corpuscular volume, MPV mean platelet volume, PC platelet count, RBCC red blood cell count. **B** Heatmap of Blockshifter enrichment z-scores of GWAS summary statistics at promoter-interacting regions (PIRs) by individual cell types using endothelial cells as a control. For each trait, comparisons have been made between each individual cell type against the control endothelial precursors. Green indicates enrichment in the labeled tissue; blue indicates enrichment in the endothelial cell control. HSC hematopoietic stem cell, CMP common myeloid progenitor, CLP common B-cell lymphoid progenitor, MK megakaryocytes, Mon monocytes, Ery erythroblasts, nB naive B cell, nCD4 naive CD4+ cells, nCD8 naive CD8+ cells.

regulatory elements that mirror the cell-type specificity of the interacting gene's expression.

We then applied chromatin assortativity analysis[37], which recognizes the preference of network's nodes to attach to others that have similar features, to test the potential of liCHi-C to discover proteins or chromatin marks mediating genomic contacts within the nucleus (Fig. 3C and Supplementary Fig. 6A). We observed that genomic regions enriched in H3K9me3 histone modification, which have been associated with constitutive heterochromatin and lamina-associated domains, are highly interconnected and may form topological hubs that collaborate with epigenetics to promote gene silencing (Fig. 3D and Supplementary Fig. 6B–D). Collectively, these results illustrate the power of liCHi-C to suggest, after further functional validation, aspects on the diversity of factors and mechanisms regulating genome architecture.

## liCHi-C enables the discovery of disease-relevant cell types and disease-associated genes and pathways

Genetic variation, which frequently affects the noncoding genome, occurs at various levels ranging from single-nucleotide variants, such as single-nucleotide polymorphisms (SNPs), to larger structural variants (SVs). To test liCHi-C's ability to uncover associations between noncoding SNPs and disease etiology, we integrated summary statistics from 39 genome-wide association studies (GWAS), including seven autoimmune diseases, eight myeloid cell traits, four lymphoid cell traits, nine blood malignancies and eleven traits non-related to the hematopoietic hierarchy (Supplementary Data 4). Using Blockshifter analysis we showed that PIRs called by liCHi-C in a cell type, independently of whether these are shared or not across cell types, are enriched for genetic variants associated with traits or diseases relevant to the cell type (Fig. 4A, B). For instance, variants associated with the final maturation of myeloid cells tend to be more enriched at PIRs in

mature myeloid cells. Interestingly, similar enrichment profiles were obtained by liCHi-C and PCHi-C despite the dramatic reduction in starting material (Supplementary Fig. 7A). These data demonstrate that liCHi-C can trace the ontogeny of activity of the noncoding genome in association with pathogenic traits and enables the identification of cell types presumably implicated in disease etiology.

We next used the Bayesian prioritization strategy COGS to rank putative disease-associated genes based on GWAS and liCHi-C data. Excluding SNPs at promoter or coding regions, we assigned 24,504 distal noncoding SNPs to potential target genes, which were located at a median genomic distance of 187 kb (Fig. 5A). Remarkably, only 19.58% of these were linked to the nearest gene and 38.16% potentially controlled more than one gene. These results highlight the importance of being able to generate data on long-range interactions between promoters and regulatory elements to avoid misleading associations based on proximity in the context of gene regulation and disease.

Specifically, using this computational framework on liCHi-C data we prioritized 6230 candidate genes (with a median of 134 genes per trait/disease at gene-level score >0.5) and 56 candidate gene pathways according to the Reactome Pathway Database[38] (Fig. 5B). These genes were highly similar to those prioritized by PCHi-C using 40 M cells (Supplementary Fig. 7B, C). For instance, our data suggested that deregulation of *TLE3*, a gene that encodes for a co-repressor protein that negatively regulates canonical *WNT* signaling[39], could confer susceptibility to lymphocytic leukemia (Fig. 5C). Moreover, it also pointed out "activation of BH3-only proteins" pathway, which is involved in the canonical mitochondrial apoptosis[40], as being potentially implicated in this type of cancer (Fig. 5B). These and many other examples of expected and unexpected genes and pathways potentially deregulated by noncoding SNPs uncovered in our datasets, warrant follow-up studies to characterize their functional relevance in disease

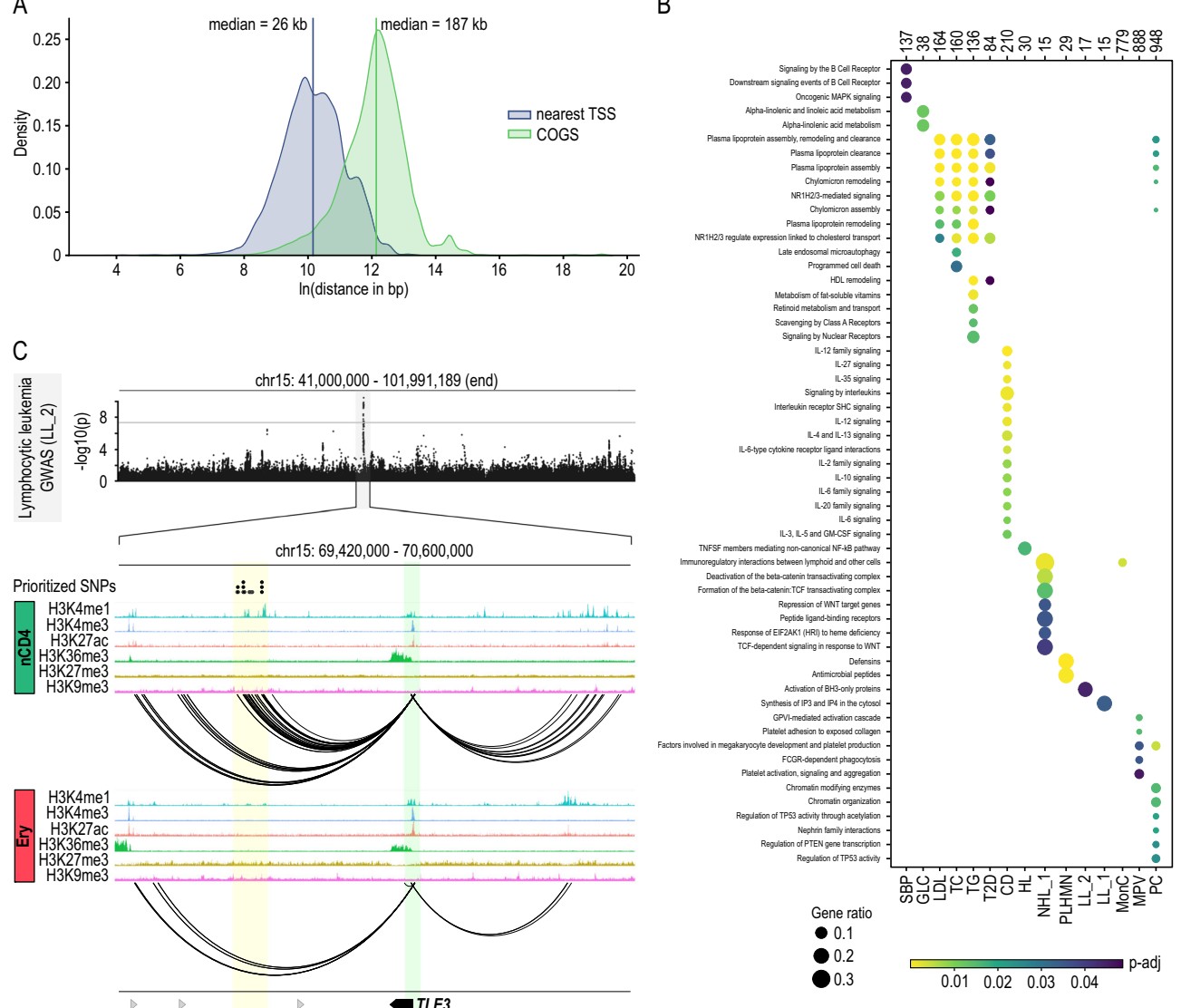

**Fig. 5 | liCHi-C enables the discovery of disease-relevant genes and gene pathways. A** Density distribution of the distance between selected SNPs and the transcription start site (TSS) of the nearest protein-coding gene (blue) and the TSS of the gene prioritized by COGS (green). For each GWAS trait, we selected all the significantly associated SNPs ($P$ value $<5 \times 10^{-8}$) interacting with at least one protein-coding gene prioritized by COGS. Coding SNPs or noncoding SNPs overlapping promoters were excluded. Vertical lines represent median distance. **B** Bubble plot of traits with significant enrichment ($P$-adj <0.05) in one or more pathways from the Reactome database. Top numbers indicate the total number of genes analyzed for each trait (gene score >0.5), bubble size indicates the ratio of test genes to those in

the pathway, and blue to yellow corresponds to decreasing adjusted $P$ value for enrichment. $P$ values were calculated using a hypergeometric distribution test and adjusted for multiple comparison using a false discovery rate (FDR) cutoff of 0.05. **C** Example of prioritized gene (*TLE3*) for the trait lymphocytic leukemia (LL_2) in naive CD4$^+$ cells. Top: Manhattan plot; gray line indicates the significance cutoff ($5 \times 10^{-8}$). Middle: prioritized SNPs (black dots) at naive CD4$^+$ (nCD4)-specific enhancer. Bottom: *TLE3* regulatory landscape nCD4 and erythroblasts (Ery). Green shade depicts the gene promoter, while yellow shades depict putative enhancer region for that gene enriched in prioritized noncoding SNPs. Arrows symbolize gene placement and orientation along the genomic window.

phenotypes. Nonetheless, our results demonstrate the power of liCHi-C to identify potential disease-causative genes and pathways.

## liCHi-C can be used to simultaneously diagnose and discover translocations, copy number variations (CNVs) and topological alterations in tumoral samples

After demonstrating the capacity of liCHi-C to prioritize noncoding SNPs with potential functional relevance in clinical settings, we sought to investigate SVs affecting larger genomic regions, including translocations and CNVs. Most of the ligation events detected by proximity-ligation methods, such as liCHi-C, occur between sequences in proximity along the linear genome and the frequency of these events decreases logarithmically with the genomic distance that separates them. However, a translocation alters the linear genome and artificially

increases the number of ligation events between the juxtaposed regions. Based on this, we reasoned that liCHi-C could detect genome-wide chromosomal translocations, identify the breakpoints and uncover alterations in gene promoter interactions that could shed light on the pathogenic role of SVs. To test this hypothesis, we generated high-quality liCHi-C libraries using primary blasts from two pediatric B-cell acute lymphoblastic leukemia (B-ALL) samples (500k cells per library) (Supplementary Data 1 and Supplementary Fig. 8B–E) previously analyzed by routine clinical assays (Fig. 6A and Supplementary Fig. 8A). According to FISH and karyotyping analysis, B-ALL sample 1 carried a balanced translocation between chromosomes 8 and 14, which appeared as "butterfly" blocks of interactions between the translocated chromosomes on the liCHi-C interaction matrix (Fig. 6A, top). A closer examination of the interaction directionalities

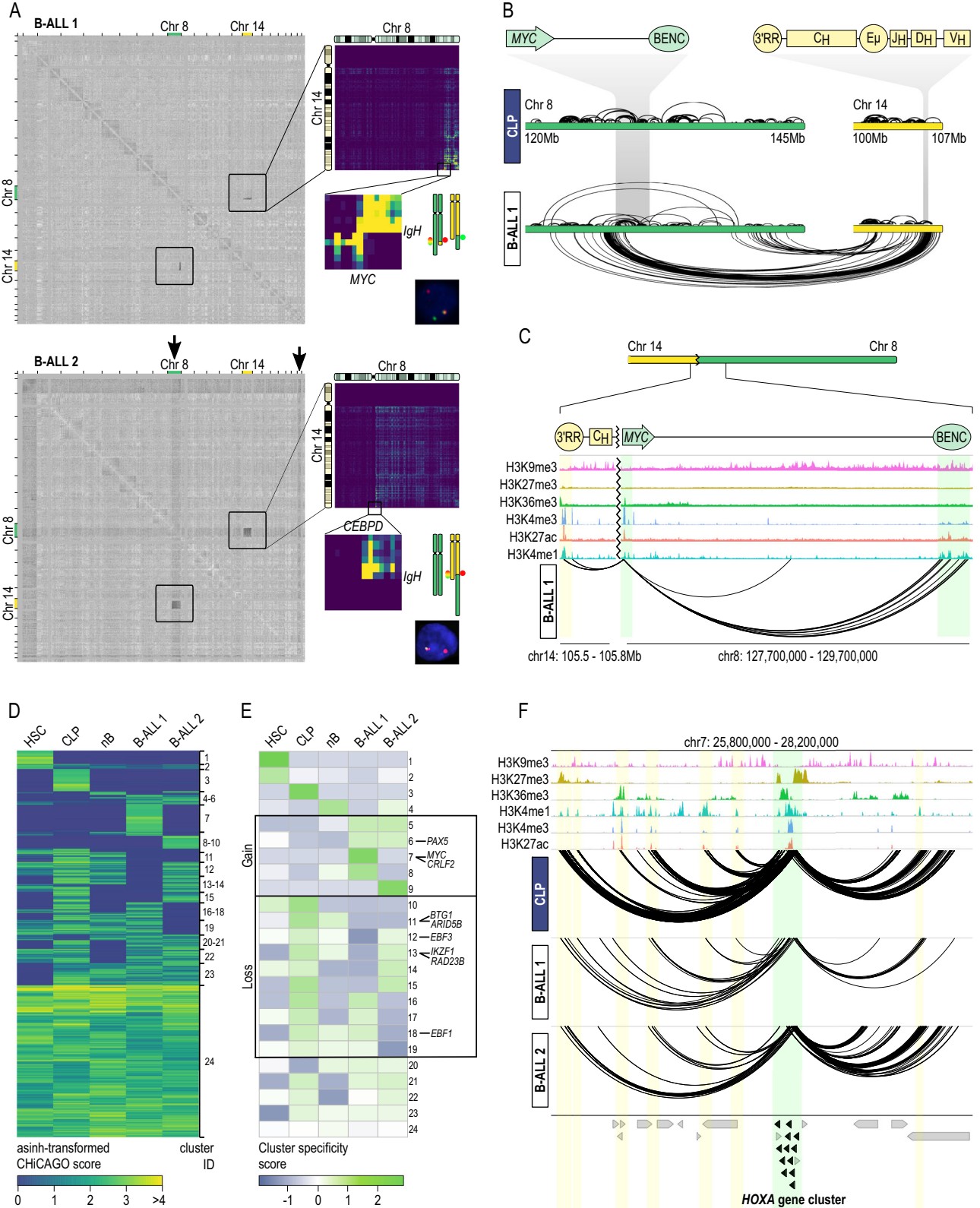

identified the restriction fragments affected by the breakpoints and allowed the reconstruction of the resulting chromosome where the promoter of *MYC* becomes rearranged next to the "constant" gene cluster of the IgH locus (Fig. 6B). Through this focused analysis of promoter interactions, our data suggested that *MYC* expression may be simultaneously controlled by the "*MYC* blood enhancer cluster" (BENC)[41], located 2 Mb downstream, and the IgH-specific enhancer 3′-regulatory region (3′-RR)[42], rearranged 300 kb upstream (Fig. 6C).

B-ALL sample 2 carried an unbalanced translocation between the same chromosomes that generated single blocks of contacts on the normalized liCHi-C interaction matrix (Fig. 6A, bottom). Consequently, the *CEBPD* gene[43] becomes juxtaposed to the IgH locus, which potentially alters its regulatory landscape (Supplementary Fig. 8F, G). Collectively, these results demonstrate liCHi-C's capacity to generate high-quality chromatin conformation maps and regulatory landscapes directly from primary patient tissue samples to detect any type of

**Fig. 6 | liCHi-C simultaneously detects translocations, copy number variations, and topological alterations in tumor samples. A** Detection of structural variants using liCHi-C data for B-cell precursor acute lymphoblastic leukemia (B-ALL) 1 (top) and B-ALL 2 (bottom). Gray matrices on the left represent the log2 ratio between B-ALL and CLP contact matrices at 1 Mb resolution across the genome. Black arrows indicate the location of copy number gains. On the top right: matrices at a 250 kb resolution of the chromosomes involved in the translocations. On the middle right: zoom-ins of the breakpoint regions and schematic representation of the translocated chromosomes and the location of the FISH probes. On the bottom right: FISH images displaying the translocation. **B** Top: Schematic representation of *MYC* and *IgH* genes loci. Bottom: interaction landscape of chromosomes 8 and 14 in common B-cell lymphoid progenitors (CLP) and B-ALL 1 sample. Interactions within and between chromosomes are represented over and below the chromosomes,

respectively. **C** Reconstruction of *MYC* promoter interaction landscape on the derivative chromosome in B-ALL 1. Green shades depict the *MYC* promoter and BENC enhancer from chromosome 8, and the yellow shadow the 3' RR enhancer from chromosome 14. **D** Heatmap of asinh-transformed CHiCAGO score of significant interactions in at least one cell type clustered using Autoclass algorithm. HSC hematopoietic stem cell, nB naive B cell. **E** Heatmap of cluster specificity score of each Autoclass cluster. Clusters containing interactions specific to B-ALL or interactions specifically lost in B-ALL are boxed. **F** *HOXA* gene promoter-centered interaction landscape (arcs) in CLP, B-ALL 1, and B-ALL 2 samples. Green shade depicts the *HOXA* gene cluster, while yellow shades depict putative enhancer regions for that gene in any of these cell types. Arrows symbolize gene placement and orientation along the genomic window.

translocation and surmise the pathogenic effects at a genome-wide scale.

In addition to chromosome rearrangements, we tested whether liCHi-C can be used to detect CNVs. Gains or losses of genetic material imply an increase or decrease in the ligation events within the altered regions, respectively. Therefore, we reasoned that CNVs should appear when comparing liCHi-C normalized interaction matrices of cells carrying the CNV against control cells. To test this hypothesis, we focused on the B-ALL sample 2 that, according to the karyotyping analysis, carried a trisomy of chromosome 21 and a partial trisomy affecting the translocated region of the q arm of chromosome 8. As shown in Fig. 6A bottom, both trisomies were identified, demonstrating that liCHi-C can be used for genome-wide detection of CNVs and to scan breakpoints from primary patient tissues without the need for a reference.

Finally, we tested the ability of liCHi-C to identify disease-specific regulatory 3D chromatin landscapes that may be implicated in disease etiology. To do so, we applied Autoclass Bayesian clustering of liCHi-C significant interactions (CHiCAGO score >5) called either on B-ALL samples or on the postulated healthy cells of origin of these hematological malignancies (i.e., HSC, CLP, and naive B cells) (Fig. 6D). Specificity score analysis of each cluster in each cell type identified promoter interactions specifically acquired (C5-9) or lost (C10-19) in one or both B-ALL samples (Fig. 6E), which included key transcription factors involved in B-cell differentiation and function (e.g., *PAX5*[44], *ARID5B*[45]), well-known tumor suppressor genes (e.g., *BTG1*[44], *IKZF1*[44]) and protooncogenes (e.g., *MYC*[46]). For instance, several *HOXA* genes have been associated with normal hematopoiesis and blood malignancies[47,48]. According to liCHi-C data, these genes lose connectivity with their enhancers, which could link their transcriptional deregulation with malignant transformation (Fig. 6F). Collectively, these results support the broad applicability of liCHi-C to uncover factors and mechanisms involved in disease etiology through simultaneously identifying disease-specific promoter-centered genome topologies and detecting translocations, CNVs, breakpoints and their effects on transcriptional deregulation.

### liCHi-C can be customized to improve its resolution

liCHi-C resolution is determined by the restriction enzyme used for the library generation and defines the range of significant interactions to be detected. To demonstrate the adaptability of our method to interrogate promoter interactions at different resolutions, we used primary blasts from a third B-ALL sample to generate two high-quality liCHi-C libraries using a six-cutter restriction enzyme (HindIII) and a four-cutter restriction enzyme (MboI) respectively (~250k cells per library) (Supplementary Data 1 and Supplementary Fig. 8H–K). Patient 3 carried a monosomy of chromosome 7, which was clearly identified as a reduction of reads on the contact matrix generated by both restriction enzymes (Fig. 7A, B).

liCHi-C libraries generated with MboI detected 1.78 times more significant interactions (Supplementary Fig. 8J), which were

characterized by having half of the median linear distance between promoters and their interacting regions (Fig. 7C). Indeed, although the shortest significant interaction was similar for both libraries (2574 bp for HindIII and 1939 for MboI), the highest frequency of interactions was found at a distance 2.15 times larger for HindIII restriction (Fig. 7C). These data, illustrated by the promoter interactome of the *DDX41* (Fig. 7D), a DEAD box RNA helicase associated with B-ALL and other blood malignancies[49], demonstrates that the use of a four-cutter restriction enzyme increases the power to detect short-range interactions and compromises the detection of the long-range ones. Taken together, our results demonstrate liCHi-C capability to provide fundamental and clinical insights about gene-regulatory interactions at different levels of resolution.

## Discussion

High-throughput chromatin conformation capture methodologies such as Hi-C[50] have revolutionized our understanding of long-range gene transcriptional control. However, many aspects of its dynamics along in vivo differentiation and stimulation, as well as its alteration in disease, remain largely unexplored due to the lack of genome-wide methodologies to study the promoter-centric genome architecture at high resolution with low-input material. Whereas single-cell[51,52] or low-input[24,26,53,54] approaches exist, these generate sparse contact maps with low resolution that do not allow the study of specific chromatin interactomes. More recently, HiChIP[55] and HiCuT[56] methods have been developed to study long-range chromatin interactions mediated by a specific protein. Although both technologies are compatible with low-input cell numbers, these rely on the availability of high-quality antibodies that recognize the target protein. Besides, these methods cannot be used to compare chromatin interactomes between conditions in which the binding of the target protein is different, which is very common due to the inherent dynamic nature of chromatin.

To overcome these limitations, we have developed liCHi-C, a mini-input cost-effective method to robustly map and compare promoter interactomes at high resolution in rare cell populations previously unmeasurable. Up to 12 liCHi-C libraries can be generated in 6 days with a total cost of 1500 euros per library (including sequencing cost). Unlike methods that depend on enrichment based on the use of antibodies[55–58], liCHi-C only relies on biotinylated RNAs designed to hybridize against the annotated promoters to ultimately enrich for promoter interactions from a Hi-C library. Thus, it is able to identify long-range contacts of both active and inactive promoters and robustly compare interactomes between any condition. In addition, this capture strategy provides high versatility since any customized capture system from a wide range of coverage can be designed according to the interactome to be studied. For instance, liCHi-C can be easily coupled with capture systems designed to study the interactome of a collection of noncoding alterations or the enhancer interactome. Indeed, it can even be adapted to the study of the interactome of just a few loci as other 3C-based capture methods do[59,60].

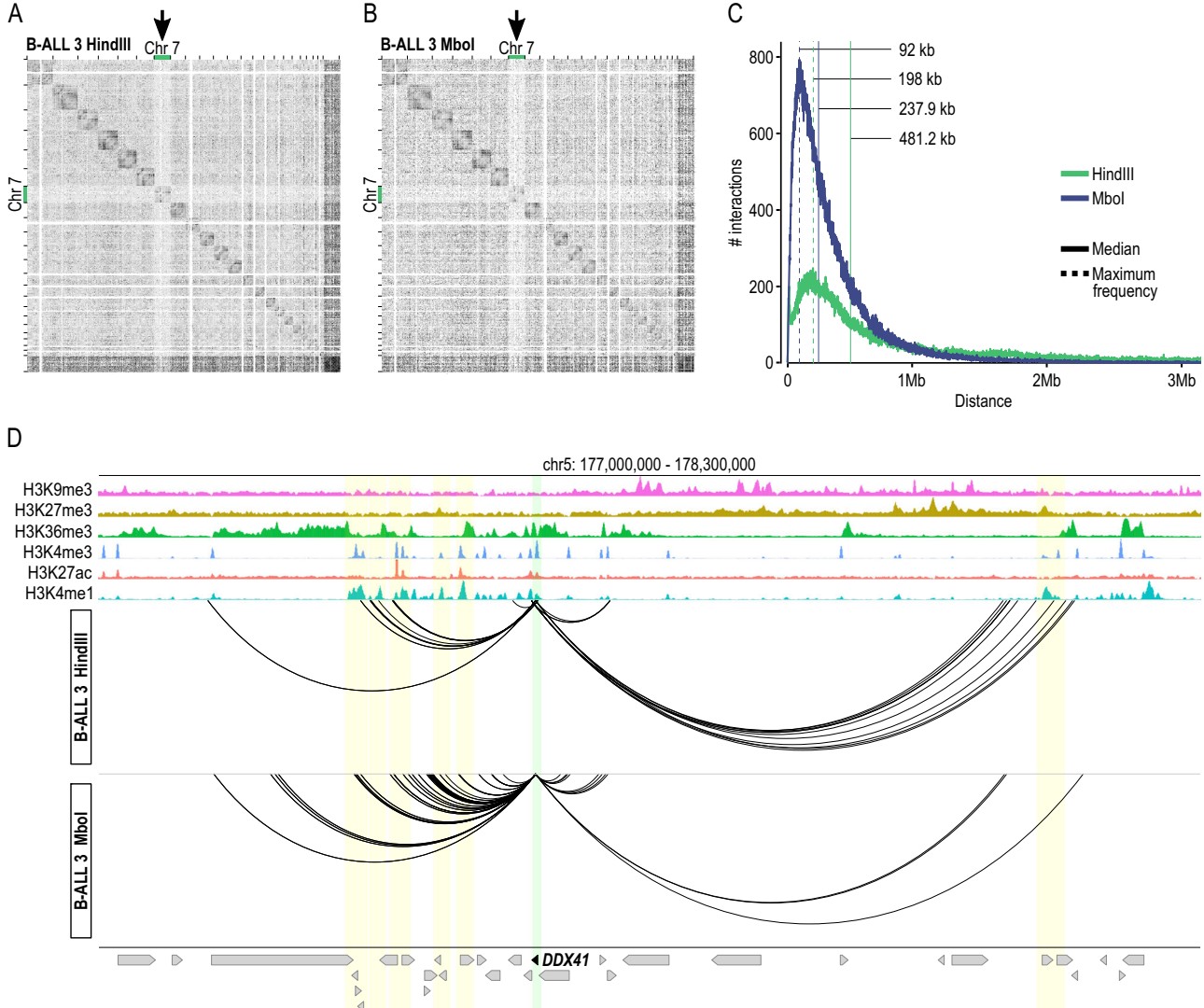

**Fig. 7 | Comparison of liCHi-C libraries at different resolutions.** Detection of structural variants using liCHi-C data for B-cell precursor acute lymphoblastic leukemia (B-ALL) sample 3 generated using a six-cutter restriction enzyme (HindIII) (**A**) and a four-cutter restriction enzyme (MboI) (**B**). Gray matrices represent the ratio between B-ALL and CLP contact matrices at 1 Mb resolution across the genome. Black arrows indicate the location of copy number losses. **C** Absolute frequency of interaction according to the genomic distance between interacting regions for liCHi-C libraries generated with MboI (blue) and HindIII (green) digestion, respectively. The median and highest absolute frequency of interactions are represented by solid and dashed lines, respectively. **D** *DDX41* gene promoter-centered interaction landscape (arcs) generated with HindIII (top) and MboI (bottom), respectively. Green shade depicts the *DDX41* gene promoter, while yellow shades depict putative enhancer regions for it. Arrows symbolize gene placement and orientation along the genomic window.

liCHi-C significantly broadens the capacity for studying organism developments, in vivo cell commitment, and cellular response to a wide range of external stimuli. As a proof of concept, we have used liCHi-C to study human hematopoietic hierarchy. Our data demonstrates that the promoter interactome can identify the differentiation trajectory. In addition, this suggests a massive dynamic rewiring of the three-dimensional epigenetic landscape parallel to transcriptional decisions during in vivo cell commitment.

liCHi-C enables the study of primary samples, thus addressing a key limitation of PCHi-C to apply this type of analysis to clinical samples. This method is especially relevant as most inherited and acquired mutations and epimutations for common human diseases, which largely remain unexplored, are all highly enriched at regulatory elements and cannot be readily modeled in in vitro systems. Genetic and epigenetic alterations at distal regulatory elements have the potential to alter the regulatory properties and ultimately lead to quantitative changes in the expression of distal target genes with pathological outcomes. As distal regulatory elements and their topological properties are highly dynamic, cell-type specific, and state-dependent, it is critical to identify the relevant human cell types for each disease and profile their full repertoire of regulatory elements and target genes. Here, we demonstrate that liCHi-C is able to fulfill this need. Focused on GWAS data, we have demonstrated that liCHi-C identifies unexpected etiological associations and exposes disease-associated genes and pathways. Although we have mostly focused on inherited risk factors, our computational framework can be adapted to study acquired mutations and epimutations at any type of distal regulatory elements (e.g., silencer or primed enhancer), since liCHi-C identifies long-range contacts promoters independently of their activity.

How chromatin organization contributes to disease pathogenesis remains largely unexplored. As we have shown, liCHi-C has the potential to contribute to filling this gap of knowledge. Taking malignant neoplasms as a model, we have shown that liCHi-C can be used to discover disease-specific topological alterations in clinical samples and generate hypotheses about genetic factors underlying disease

mechanisms. Simultaneously, liCHi-C can uncover SVs at a genomic scale, resolve the positions of their breakpoints and predict their functional effects, including the formation of regulatory landscapes, in an agnostic manner. Although we focused on the analysis of translocations and duplications associated with cancer, liCHi-C is also able to identify and characterize other types of SVs, such as inversions and deletions in any disease context. In addition, the liCHi-C method holds high potential for disease diagnosis equivalent to other chromosome conformation capture technologies[61,62], since SVs are hallmarks of mental retardation, infertility, developmental disorders and cancer.

Despite the broad applicability of our method, there are still several factors to take into consideration. One inherent limitation of liCHi-C is that its resolution is determined by the restriction enzyme used for the library generation. However, as we have demonstrated, it can be increased by replacing the enzyme by another one with greater resolution or exchange it for micrococcal nuclease. In addition, the interpretation of the biological meaning of noncoding alterations purely based on distal chromatin interactions can be challenging. Nevertheless, an integrative analysis of liCHi-C data, gene expression and chromatin states might be indicative of causal relationships, which should be validated with functional assays. Finally, the identification of exact genomic coordinates of the SV breakpoints is not possible unless these map near restriction sites. However, long-range sequencing approaches coupled with liCHi-C can allow to map the SVs at the nucleotide resolution. Besides, this combinatorial approach increases the mapability efficiency of the chromatin contacts at repetitive sequences, which can be target of SVs and other mutations and epimutations. Despite these considerations, our data proves that it is feasible to generate high-quality genome-wide promoter interaction maps from low amounts of primary patient material. We anticipate that the robustness and the inherent flexibility for customization make liCHi-C an attractive option that will allow the analysis of spatial genome architecture within reach of personalized clinical diagnostics and development biology.

## Methods

### Cell isolation

Naive B cells (nB), naive CD4$^+$ cells (nCD4), naive CD8$^+$ cells (nCD8), and monocytes (Mon) were obtained from peripheral blood mononuclear cells (PBMCs) from venous blood following standard BLUEPRINT protocols. Specifically, nB, nCD4 and nCD8 were isolated using STEMCELL Technologies Enrichment kits (cat. #19254, #19309, and #19158, respectively). Mon were isolated by Miltenyi Biotec kit (cat. #130-091-765). CD34 + cells isolated from cord blood mononuclear cells were selected with the human CD34 Miltenyi Biotec kit (cat. #130-046-702) and in vitro differentiated into Megakaryocytes (Mk) and Erythroblasts (Ery). Specifically, thrombopoietin and IL1β for 10 days and erythropoietin, SCF and IL3 for 14 days were used to differentiate Mk and Ery, respectively. Hematopoietic stem and progenitor cells (HSC; CD34$^+$, CD38$^-$) common myeloid progenitors (CMP; CD34$^+$, CD38$^+$, CD33$^+$) and common lymphoid B-cell progenitors (CLP; CD34$^+$, CD38$^+$, CD19$^+$) were purified by Miltenyi Biotec CD34-positive selection kit (cat. #130-046-703) and fluorescence-activated cell sorting (FACS) from four donations (two per biological replicate) of 15-to 22-week-old human fetal liver and fetal bone marrow as previously described[63]. Briefly, after CD34$^+$ selection, positive cells were stained for flow cytometry with the following fluorophore-conjugated monoclonal antibodies, all from BD Biosciences, and used at the manufacturer's recommended concentration: CD34 PECy7 (cat. #348811; 1:20 dilution), CD38 FITC (cat. #555459; 1:5 dilution), CD19 BV421 (cat. #562440; 1:20 dilution) and CD33 APC (cat. #551378; 1:5 dilution). FACS was performed using a BD FACSAria Fusion with BD FACSDiva 8.0.2 software. For more detail about sorting strategies, see Supplementary Fig. 4A. B-ALL samples were isolated from bone marrow of pediatric patients after CD19$^+$ selection using FACS sorter. All samples were obtained from consent donations of volunteers after approval by the Ethics committee.

### Fixation and cell quantification

Cells were quantified, resuspended in 1 ml of RPMI 1640 culture medium containing 10% FBS and 2% of methanol-free formaldehyde (Thermo Fisher cat. #28908) and incubated in a rocker for 10 min at room temperature. Formaldehyde was quenched with glycine to a final concentration of 0.125 M. Then, cells were washed with cold 1× PBS. Specific cell numbers were sorted into low-retention 1.5-ml tubes (BD FACSJazz sorter, 1.0 Drop Pure sorting mode), pelleted, flash-frozen in dry ice and stored at −80 °C.

### liCHi-C method

During the whole protocol, low-retention tips and tubes were used to minimize cell loss. Pelleted cells were softly resuspended in 500 μl ice-cold lysis buffer (10 mM Tris-HCl pH 8.0, 10 mM NaCl, 0.2% IGEPAL CA-630, 1× cOmplete EDTA-free protease inhibitor cocktail (Merck cat. #11873580001)) and incubated 30 min on ice to extract the nuclei. Soft inversions of the tube were performed during the incubation every 5 min. Nuclei were centrifuged (1000×$g$ and 4 °C for 10 min) and 450 μl of supernatant was discarded (50 μl of supernatant was left in the tube to avoid cell losses). The cell pellet was resuspended in 500 μl ice-cold 1.25× NEB2 buffer (New England Biolabs cat. #B7002S). After centrifugation (1000×$g$ and 4 °C for 10 min), 500 μl of supernatant was removed.

Afterward, 129 μl of 1.25× NE Buffer 2 were added to the low-retention tube to obtain a final volume of 179 μl. Then, 5.5 μl of 10% SDS (AppliChem cat. #A0676,0250) were laid on the wall of the tube and mixed by inversion. After incubation at 37 °C and 950 rpm for 30 min, 37.5 μl 10% Triton X-100 (AppliChem cat. #A4975,0100) were laid on the wall of the tube, mixed by inversion and incubated at 37 °C and 950 rpm for 30 min. Chromatin within the nuclei was overnight digested at 37 °C and 950 rpm after adding 7.5 μl of HindIII restriction enzyme at 100U/μl or 37.5 μl of MboI restriction enzyme at 25U/μl (New England Biolabs cat. #R0104T or #R0147M). The following day, an extra digestion during one more hour was performed after adding 2.5 μl of the HindII enzyme or 12.5 μl of the MboI enzyme.

After digestion, MboI enzyme was washed with NEBuffer 2 by centrifuging the sample and removing the supernatant to avoid possible re-digestion of ligated fragments afterward. Cohesive restriction fragment ends were filled in during 75 min at 37 °C. To do so, 30 μl of master mix composed by 5 μl of 5U/μl Klenow polymerase (New England Biolabs cat. #M0210L), 0.75 μl of each 10 mM dCTP, dGTP, and dTTP, and 18.75 μl of 0.4 mM biotin-14-dATP (Invitrogen #19524-016) were added.

In-nucleus ligation of DNA fragments was carried out during 4 h at 16 °C after adding 12.5 μl of 1U/μl T4 DNA ligase (Thermo Fisher cat. #15224025), 50 μl of 10× ligation buffer (NEB #B0202S), 5 μl of 10 mg/ml BSA (NEB # B9001S) and 170.5 ml of water. Afterward, DNA ligation products were decrosslinked by adding 30 μl of Proteinase K 10 mg/ml (Merck cat. #3115879001) and incubating overnight at 65 °C. The following day, an extra decrosslink during two more hours was performed after adding 15 μl of the Proteinase K enzyme.

To purify the decrosslinked DNA ligation products, a single phenol−chloroform−isoamyl alcohol (25:24:1 v/v) purification was carried out followed by ethanol precipitation for 1 h at −80 °C in the presence of 30 μg Glycoblue (Thermo Fisher cat. #AM9515) as a coprecipitant. DNA ligation products were resuspended in 130 μl of nuclease-free water and concentration was assessed by fluorimetric quantification using the Qubit dsDNA HS Assay Kit (Thermo Fisher cat. #Q32851).

Optional 3C controls assessing the detection of cell-type invariant interactions in the HindIII liCHi-C libraries can be performed by amplifying 50−100 ng of the DNA with 37 cycles of PCR amplification (see Supplementary Data 5 for primer information) and running the

reactions in a 1.6% agarose gel. Correct fill-in and ligation can also be tested by reamplifying 2.5 μl of PCR products five more cycles, differentially digesting the product with either HindIII, NheI (Thermo Fisher cat. #ER0975), both enzymes or none and running the product on a 1.6% agarose gel.

Biotin removal of the non-ligated ends was skipped. DNA ligation products were sonicated using Covaris M220 focused-ultrasonicator (20% duty factor, 50 peak incident power, 200 cycles per burst, 65 s) in 130 μl tubes (Covaris cat. #520077). After shearing, DNA ends were repaired by adding 6.5 μl of T4 DNA polymerase 3U/μl (New England Biolabs cat. #M0203L), 6.5 μl of T4 polynucleotide kinase 10U/μl (New England Biolabs cat. #M0201L) 1.3 μl of Klenow polymerase 5U/μl (New England Biolabs cat. #M0210L), 18 μl of dNTP mix 2.5 mM each and 18 μl of 10× ligation buffer (New England Biolabs cat. #B0202S) and incubating for 30 min at 20 °C.

Biotinylated informative DNA ligation products were pulled down using Dynabeads MyOne streptavidin C1 paramagnetic beads (Thermo Fisher cat. #65001). After thorough washing of the ligation products-beads complex and having the sample in 35.7 μl of volume, blunt DNA fragments on the beads were adenine-tailed by adding 7 μl of Klenow 3′→5′ exo- polymerase 5U/μl (New England Biolabs cat. #M0212L), 2.3 μl of dATP 10 mM and 5 μl NEB2 of 10x NEBuffer 2 and incubating the mixture 30 min at 37 °C and a further 10 min at 65 °C to inactivate the enzyme.

After thorough washing of the ligation products-beads complex and having the sample in 50 μl of 1× ligation buffer, PE Illumina adapters (Supplementary Data 5) were ligated to the adenine-tailed DNA fragments by adding 1 μl of T4 DNA ligase 2000U/μl (New England Biolabs cat. #M0202T) and 4 μl of preannealed adapter mix 15 μM and incubating the mixture 2 h at room temperature.

The bead-bound ligation products were amplified 8–13 cycles by PCR (Supplementary Data 5 for cycle recommendations according to starting cell number) using Phusion high-fidelity PCR master mix with HF buffer (New England Biolabs cat. #M0531L).

After recovering the amplified library from the supernatant, size distribution was tailored to 300–800 bp by double-sided size selection and purified using CleanNGS SPRI beads (0.4–1 volumes; CleanNA cat. #CNGS-0050). DNA concentration was quantified on an Agilent Tapestation platform using high sensitivity D1000 ScreenTape system, and samples were stored at −20 °C.

Enrichment of promoter-containing ligation products was performed using SureSelectXT Target Enrichment System for the Illumina Platform (Agilent Technologies) as instructed by the manufacturer, and the library was amplified four cycles by PCR using Phusion high-fidelity PCR master mix with HF buffer (New England Biolabs cat. #M0531L). Finally, the end product was purified using CleanNGS SPRI beads (0.9 volumes; CleanNA cat. #CNGS-0050) and paired-end sequenced.

## Sequencing
B-ALL liCHi-C libraries were sequenced by Macrogen Inc using HiseqX 150 + 150PE platform. The rest of liCHi-C libraries were sequenced by BGI Genomics using DNBseq 100 + 100PE platform.

## B-ALL cytogenetic and FISH analysis
Cytogenetic analyses of B-ALL samples were carried out on G-banded chromosomes obtained from 24-h unstimulated culture. FISH analyses were performed on fixed cell suspensions of the bone marrow using the LSI MYC probe (Metasystems, XL MYC BA) and the LSI IGH probe (Metasystems, XL IGH BA), respectively. Between 200 and 400 interphase nuclei were scored.

## liCHi-C processing
Paired-end reads were processed using HiCUP[22] (0.8.2). First, the genome was computationally digested using the target sequence of the restriction enzyme. Then, the different steps of the HiCUP pipeline

were applied to map the reads to the human genome (GRCh38.p13), filter out all the experimental artifacts and remove the duplicated reads and retain only the valid unique paired reads. To assess the capture efficiency, we filtered out those paired reads for which any end overlaps with a captured restriction fragment, retaining only the unique captured valid reads for further analysis. Library statistics for all samples are presented in Supplementary Data 1.

## liCHi-C interaction calling
Interaction confidence scores were computed using the CHiCAGO R package[23,64] (1.14.0). In summary, this pipeline implements a statistical model with two components (biological and technical background), together with normalization and multiple testing methods for capture Hi-C data. CHiCAGO analysis was performed in merged samples to increase the sensitivity, after assessing for reproducibility between biological replicates using: (i) the stratum-adjusted correlation coefficient according to ref. [65], averaged over chromosomes, (ii) principal component analysis, and (iii) hierarchical clustering. The reproducibility score was also used to compare PCHi-C and liCHi-C libraries of the same cell types. Significant interactions with a CHiCAGO score ≥5 were considered as high-confidence interactions. Interaction statistics for all samples are presented in Supplementary Data 1. liCHi-C datasets are available in EGA under the accession number EGAS00001006305.

## Loop calling using Hi-C, low-C, and TagHi-C
Loop calling was performed on Hi-C, TagHi-C, and Low-C data. Accession information is presented in Supplementary Data 2. Datasets were processed with HiCUP[22] (0.8.2) as detailed above, using the human (GRCh38.p13) and mouse reference genomes (GRCm39). Loop calling was performed with three different methods: HICCUPS[27] (1.22.01), Mustache[28] (1.2.7), and HiCExplorer[29] (3.7.2). All loop callers were used with default parameters on Knight–Ruiz normalized matrices at 5 kb resolution and a maximum loop distance of 8 Mb. Specific parameters for HICCUPS were: -cpu-ignore_sparsity; for Mustache, the P value threshold was set to 0.05.

## Clustering of promoter interactions
Interactions were clustered using the Autoclass (3.3.6) algorithm and for each cluster a specificity score was computed using the asinh-transformed CHiCAGO scores[5]. Clustering of cell types was performed using a hierarchical method with average linkage based on Euclidean distances, and principal component analysis was performed using the prcomp function in R. For interaction data handling we used an in-house R package which allows us to compute distance distributions, filter interactions by the presence of histones marks, or generate virtual 4C of specific genes.

## ChIP-seq processing
Paired-end reads were processed following ENCODE standards. Reads were trimmed using Trim Galore (0.6.5) to remove sequencing adapters, and then mapped using bowtie2 (2.3.2) to the reference genome (GRCh38.p13) with -very sensitive parameter. We filtered out low-quality reads, reads overlapping the ENCODE blacklist and duplicated reads. Peak calling was performed using macs2 (2.2.7.1) in broad and narrow mode depending on the histone mark using an input sample as control, with default parameters. Bigwig files were generated using the function bamCoverage from deepTools (3.2.1) and scaled based on the background normalization of the samples before merging the biological replicates together for visualization purposes[66]. All ChIP-seq data analyzed in this article are presented in Supplementary Data 3, which also includes the accession information.

## Histones mark enrichment at PIRs
Enrichment of the different histones mark ChIP-seq in both promoters and npPIRs was performed using a permutation test

implemented in RegioneReloaded (1.0.0) (https://github.com/RMalinverni/regioneReloaded) with 5000 randomizations using the randomizeRegions option. Promoter interactions were classified based on the RPKM expression quartiles of their genes, and the promoters and npPIRs analyzed separately, generating four groups of either promoters or npPIRs from lower expression to higher expression.

## Chromatin assortativity

Chromatin assortativity was computed for the promoter−pPIR and promoter−npPIR subnetworks separately using ChAseR (0.0.0.9) R package (https://bitbucket.org/eraineri/chaser)[37]. We validated our results against a set of 1000 randomizations preserving the genomic distances between nodes and the chromosomes distribution of interactions. The abundance of each histone mark on the nodes of the networks was also computed.

## Gene ontology and pathways enrichment analysis

Gene ontology (GO) enrichment analysis was performed using clusterProfiler (4.2.2) R package for the three ontologies: molecular functions, biological processes and cellular components. The pathway analysis was done using ReactomePA[67] (1.38.0). We used the prioritized protein-coding genes by COGS (gene score >= 0.5) to compute enrichment in genes in Reactome pathways, adjusting $P$ values by false discovery rate. In both analyses, we used as universe all the genes in the capture design.

## Data visualization

To visualize the contact matrices, the bam files, containing unique captured valid reads, were transformed into pair files using bam2pairs (0.3.7) (https://github.com/4dn-dcic/pairix). Then they were converted into matrices in cool format with a resolution of 1 Mb, 250 kb and 100 kb using cooler[68] (0.8.11). The visualization of the matrices was done using HiCExplorer[69] (3.7.2). For the visualization of significant interactions generated by liCHi-C we used the WashU Epigenome Browser[70] (46.2) and karyoploteR R package[71] (1.22.0).

## GWAS summary statistics and imputation

GWAS summary data was obtained from the NHGRI-EBI GWAS Catalog[72] and from the UK Biobank - Neale Lab (UK Biobank, n.d.; http://www.nealelab.is/uk-biobank). Accession information is included in Supplementary Data 4. Those datasets that were not in GRCh38 coordinates were converted to it using *liftOver*. To avoid spuriously strong association statistics, we filtered out SNPs with $P$ value $<5 \times 10^{-8}$ for which there were no SNPs in LD ($r^2 > 0.6$ using 1000 genomes EUR cohort as a reference genotype set[73] or within 50 kb with $P$ value $< 10^5$. To increase the power of the GWAS we applied the Poor Man's Imputation to the summary statistics described above using as reference genotype set the 1000 Genomes EUR cohort[5]. We used the GRCh38 HapMap Phase II genetic map lifted from GRCh37 coordinates[74] to define regions with 1 cM recombination frequency to be used for the imputation. The MHC region (GRCh38:6:25−35 Mb) was excluded from the analysis. Manhattan plots were visualized using the qqman package[75] (0.1.8).

## GWAS enrichment at PIRs

Enrichment of SNPs in PIRs was performed using Blockshifter which considers the correlation between GWAS and PIRs[5]. Blockshifter computes a z-score using a competitive test for each trait and cell-type set.

## liCHi-C GWAS prioritizing genes

Prioritization of relevant genes for each GWAS trait and cell type was performed using the COGS algorithm[5]. Briefly, this method considers linkage disequilibrium to estimate the posterior probability of each SNP being casual for each trait. Then, these SNPs were used to compute a gene score for all the genes involved in liCHi-C significant interactions in at least one cell type. This gene score is composed of three components: coding SNPs annotated by VEP[76] (104); SNPs located in promoter regions; and SNPs overlapping Other-Ends.

## SVs analysis

Translocations of B-ALL samples were identified using PLIER[61] (0.21). Briefly, it compares the genome-wide interactions of a region against a set of random permutations computing a z-score. To detect CNVs, we applied Control-FREEC[77] (11.5), which uses a sliding window approach to calculate read count and normalize it using a control sample. Specifically, we used naive B samples as control and the following specific parameters: window size = 200,000; ploidy = 2; breakPoint Threshold = 1.3; breakPointType = 4; forceGCcontentNormalization = 2; minCNAlength = 3; mateOrientation = 0; and the hg38 mappability track. Visual inspection of both translocations and CNVs was performed by computing the ratio between the contact matrices of B-ALL and CLP samples.

## Institutional review board statement

The study design and conduct complied with all relevant regulations regarding the use of human study participants and approved by the Institutional Review Board of the Clinical Research Ethics Committee of University Hospital Germans Trias i Pujol (REF.CEI: PI-18-205). The study was conducted in accordance to the criteria set by the Declaration of Helsinki.

## Reporting summary

Further information on research design is available in the Nature Portfolio Reporting Summary linked to this article.

## Data availability

The data that support this study are available from the corresponding author upon reasonable request. All raw and processed liCHi-C datasets generated in this study have been deposited in EGA under the accession number EGAS00001006305. These datasets will be shared with controlled access in accordance with the ethical consent signed by the volunteers and it is limited to not-for-profit organizations after providing documentation of local IRB/ERB approval. Data Access Committee, led by Biola M. Javierre (bmjavierre@carrerasresearch.org), will determine access permissions in a timeframe of fifteen workdays. Reference genomes were obtained from Ensembl: GRCh38.p13 (release 104) and GRCm39 (release 106). The 1000 Genomes variants data was downloaded from the International Genome Sample Resource (data collection: 1000 Genomes on GRCh38), and the GRCh38 genetic map was obtained from http://csg.sph.umich.edu/locuszoom/download/recomb-hg38.tar.gz. PCHi-C data analyzed in the study is available at EGA (accession number: EGAS00001001911). Data used for the benchmarking of C-based methods was publicly available under the following accession codes: E-MTAB-5875 at ArrayExpress for Low-C data; EGAS00001004763 and EGAS00001001911 at EGA for Hi-C data; GSE161082 and GSE152918 at Gene Expression Omnibus (GEO) for TagHi-C data. Data of publicly available omics including ChIP-seq and RNA-seq was obtained from BLUEPRINT (http://dcc.blueprint-epigenome.eu) and ROADMAP (http://www.roadmapepigenomics.org); and GWAS Summary Statistics were downloaded from the NHGRI-EBI GWAS Catalog (https://www.ebi.ac.uk/gwas) and from the UK Biobank−Neale Lab (http://www.nealelab.is/uk-biobank). More details about access information to publicly available data used on the study (i.e., RNA-seq, ChIP-seq, GWAS, and 3C-based methods data) are included in Supplementary Data 2−4.

## Code availability

All the analyses done in the present article are accessible in the GitHub repository https://github.com/JavierreLab/liCHiC, which is linked to the https://doi.org/10.5281/zenodo.7351026.

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

## Acknowledgements

We thank Roberto Malinverni, Oliver S. Burren, Mikhail Spivakov, Vera Pancaldi, Emanuele Raineri, Sven Sewitz, and Steven W. Wingett for computational feedback. We gratefully acknowledge the participation of volunteers. We are indebted to the Barcelona Tissue Bank and the Sant Joan de Déu Hospital Biobank integrated in the Spanish Biobank Network of ISCIII for the non-fetal sample and data procurement. The human embryonic and fetal material was provided by the Joint MRC/Wellcome Trust (Grant # MR/006237/1) Human Developmental Biology Resource (http://www.hdbr.org). Finally, we also want to thank Sandra Castillo, Sergi Cuartero, Esteban Ballestar, and members of the Javierre Group for the critical discussion. We thank CERCA Programme/Generalitat de Catalunya and the Josep Carreras Foundation for institutional support. This work was supported by FEDER/Spanish Ministry of Science and Innovation (RTI2018-094788-A-I00), the European Hematology Association (4823998), the Spanish Association against Cancer (AECC) LABAE21981JAVI and PRYGN211192BUEN, the Spanish Ministry of Economy and Competitiveness PLE2021-007518 and the Carlos III Health Institute: ISCIII/FEDER PI20/00822. BMJ is funded by La Caixa Banking Foundation Junior Leader project (LCF/BQ/PI19/11690001), LT-D is funded by the FPI Fellowship (PRE2019-088005), L.R. is funded by an AGAUR FI fellowship (2019FI-B00017), P.L.-M. is funded by FPU fellowship (FPU20/03798) and O.M. is funded by an investigator fellowship from Spanish Association against Cancer (INVES211226MOLI). The funder bodies were not involved in the study design, collection, analysis, interpretation of data, the writing of this article, or the decision to submit it for publication.

## Author contributions

B.M.J. designed research; O.M., C.G., E.C., D.C., M.C., C.B., and P.M. contributed sample isolation; L.R., A.P.R., R.M., and B.M.J. performed research; L.T.-D., P.L.-M., A.N.-A., and F.S. analyzed the data; B.M.J. wrote the paper with the support of L.T.-D., L.R., A.V., and P.L.-M.; and all authors provided feedback on the manuscript.

## Competing interests

The authors declare no competing interests.
