## [Peer Review File · Nature Communications]

Reviewers' comments:

Reviewer #1 (Remarks to the Author):

This study refined the previously developed promoter-capture Hi-C method, so that it can be applied to a sample input of 50,000 cells rather than millions of cells, broadening its application to rare cell populations or clinical samples. The authors showed that the new method can discover lineage relationship among blood cell types, correlate GWAS SNPs to candidate affected genes, and identify translocations and altered chromatin interactions in leukemia samples. The data appear solid and the new method is a worthwhile addition to the existing 3D genome toolkits.

Comments:

1. It's better to call 50,000 cells as "mini-input" rather than "low input", as there are methods that can detect chromatin interactions with fewer than 1000 cells.
2. The biological findings are mostly expected or known. To demonstrate the advantages of the new method, the authors should add more references and compare it to existing C-based methods used for detecting the 3D genome of blood lineage and for detecting cancer alterations.
3. Capture-based C methods are currently not widely adopted. Can authors provide information on the cost, time expenses and adoptability of the new method?

Reviewer #2 (Remarks to the Author):

In their paper, Tomas-Daza et al. describe a variant of promoter capture Hi-C (PChi-C) suitable for low input (i.e. a minimum of 50k cells) biological samples. Multiple variants of C technology suitable for low cell numbers have emerged throughout the past years (e.g. <https://www.nature.com/articles/s41467-018-06961-0>, <https://academic.oup.com/nar/article/45/22/e184/4653540>), and this paper adds another technology to this line-up. The work is technically sound - both from wet lab and computational perspectives - and is primarily focused on benchmarking liChi-C using a set of analyses very similar to previous work from this group using PChi-C (Javierre et al. Cell 2016), with the addition of an analysis of topological alterations in two leukemia samples. While the manuscript is interesting to the part of the community that used PChi-C as this work shows that it can be applied to 250-500k cells, the technology presented here has several limitations that makes it difficult for liChi-C to compete with established methods such as Low-C and low-input Capture-C (detailed below).

specific comments:

- Supp.Fig.2a. shows that the number of significant interactions detected drops substantially in merged replicate samples when using <250k cells, with <50% interactions being detected in merged 50k replicates as compared to the two single replicates. This suggests that reproducibility takes a big hit when one applies PChi-C to relatively low cell numbers, which is reflected by separate clustering of 50k/100k samples from the rest in Fig.1c and Supp.Fig.2d. Reproducibility between replicates is assessed in the paper using an SCC correlation score, which appears to be rather high, but this doesn't seem to match with the numbers shown in Supp.Fig.2a. The authors should address this issue and further explore reproducibility at the level of individual interactions in their 50k/100k samples. For example, they should show side-by-side comparisons of biological replicate interaction tracks at selected loci (e.g. those shown in Supp.Fig.2e-g).
- In Fig.3, the authors use 500k cells for their analyses of human hematopoietic cells. While it indeed seems that liChi-C works well on 500k cells, it's much clearer (see comment above) whether it really works well on actual clinically-compatible low cell numbers, i.e. <100k. Here, competing technology

such as low-input Capture-C (<https://academic.oup.com/nar/article/45/22/e184/4653540>) appears to outcompete liChi-C, as this method produces good quality interactions profiles on as few as 10k-20k cells. The impact of liChi-C presented in its current form is therefore likely to be limited, unless the authors can substantiate liChi-C's performance in the low cell spectrum.

- A drawback of the current liChi-C protocol is that it still relies on a 6-cutter restriction enzyme (HindIII), while the entire field now used 4 bp cutter such as DpnII and MboI (or even MNase) to generate profiles with a much improved resolution. HindIII generated promoter interaction profiles are unable to pick up short-range interactions in the <20-25kb range, resulting in technology that is 'blind' to this range, which is known to frequently harbour relevant gene-regulatory interactions. Competing technologies such as low-input Capture-C all use 4-bp cutters and do not suffer from this problem. Impact of the paper would be increased if the authors could show that liChi-C can be adapted for use with 4 bp cutting restriction enzymes.

Reviewer #3 (Remarks to the Author):

Tomas-Daza et al outline the development and potential utility of their improved, low input promoter capture HiC (PCHiC, Javierre et al., Cell, 2016) method liChiC. They reveal the promoter centric 3D genome architecture of the haematopoietic lineage, overlay 3D architecture with gene expression, epigenetic marks and disease associated genetic variants all with their low input capture method.

I have a few concerns.

Major concerns

Capture HiC libraries historically have very high duplication percentages. What are the duplication rates in the samples from this work? Table 1 suggests they are perhaps as high as 90% in the 50K B cell samples but these numbers are not as apparent as the other emboldened library statistics. As duplication has a major impact on the usability and cost of these libraries, I feel that the duplication percentages should be both outlined in the relevant section of text (line 100-103) and clearly stated in Table 1.

It is also not clear from Table 1 what the final number of usable reads are per sample. This would be useful to have clearly outlined to enable a reader to determine, for example, how many reads reveal ~100,000 interactions.

The number of cells used per library for all samples is also not immediately apparent, either within the text or Table 1. For example, how many HSCs or CLPs were used for generate each library? Was it 500K? How was this decision made? How many human fetal livers were required to obtain these? Given the paper hinges on the amount of input required for the technology, not providing, or making this this information difficult to find is a significant oversight.

While I am aware the comparative analysis was performed on merged samples, I would like to see the replicates shown in Fig 2B.

While later figures show the genome-wide (if epigenetic state specific) relationship between expression and interactivity, I would like to see the impact of the differences in interactivity shown in Fig 2E-G on the expression of the specific genes shown. A simple overlay with RNA-Seq would suffice.

In the Blockshifter analysis of GWAS SNPs and PIRs Line 208 states the PIRs analyses are cell-type specific. How specific is this? For example, it was shown that CD4 and CD8+ T cells have largely overlapping 3D architectural profiles, thus should have less cell-type specific PIRs than say monocytes which possess quite unique PIR profiles. Does this mean that the analysis is underpowered to detect T cell-dependent disease profiles? Clarification would be valuable.

Minor concerns

I think many of the conclusionary statements within the paper are overstated. I include a few examples below.

Line 145 "independently of the method used or the starting number of cells (Extended Data Fig. 3f)."

1) I do not believe that the data show independence from starting cell number as all samples are either 500K (I believe, see above) or 40M. A reader could easily confuse this with the previous experiments titrating cell number. More accurate language should be used. 2). Also, the figure panels are referred to out of order.

Line 150 "demonstrated that promoter interactomes were highly reproducible and dynamic (Fig. 2b)."
– data does not show dynamics at all, I would suggest changing "dynamic" to "cell-type specific"

Line 186 : "regulatory elements for each gene in rare cell types hitherto undeterminable, and suggest that" – 'hitherto undeterminable' is overstated. These interactions may have been detected previously in HiC or other methods.

Line 197: "these results illustrate the power of liChi-C to expose new aspects on the diversity of factors and mechanisms regulating genome architecture in physiological and pathogenic settings" – the authors overlay a number of histone marks with regions of interactivity. This does not reveal factors or mechanism regulating 3D genome architecture, and certainly not in disease. It is at most an association, albeit a logical and strong one. Please reword.

Line 244: "Most of the ligation events detected by proximity-ligation methods, such as liChi-C, occur between contiguous sequences in the linear genome" – This is untrue. While it is true that the majority of ligation events occur between regions that are less than 10kB apart in the linear genome, very few relink the genome contiguously. Please reword.

Line 334: "liChi-C enables the study of primary samples, thus addressing a key limitation to apply this type of analysis to clinical samples". Please clarify that this refers to the comparison to PCHiC specifically. HiC is readily performed on primary human cells.

There are also a handful of minor grammatical errors within the manuscript.

Point-by-point responses to Reviewers' comments (NCOMMS-22-19740-T).

First, we would like to thank the reviewers for the constructive criticism and suggestions to improve the manuscript. In this revised version of the manuscript, we have addressed all the reviewer comments. In particular, upon revision, we have included:

- A more comprehensive comparison of liCHi-C against other existing C-based methods used for detecting interactomes, as suggested by reviewer #1.
- Demonstration of liCHi-C versatility to be coupled with a 4-cutter restriction enzyme to increase resolution, as suggested by reviewer #2.
- More clarifications, as suggested by reviewer #3.

To help with the structuring of this document, we have marked the reviewer's comments with **blue typography**, while our replies are shown in normal black font. Besides, the main editions to the manuscript are also marked in **blue typography** and we have included the line numbers to help the identification of these in the manuscript.

Reviewer #1 (Remarks to the Author):

This study refined the previously developed promoter-capture Hi-C method, so that it can be applied to a sample input of 50,000 cells rather than millions of cells, broadening its application to rare cell populations or clinical samples. The authors showed that the new method can discover lineage relationship among blood cell types, correlate GWAS SNPs to candidate affected genes, and identify translocations and altered chromatin interactions in leukemia samples. The data appear solid, and the new method is a worthwhile addition to the exiting 3D genome toolkits.

We thank the reviewer for highlighting the novelty of our method and the quality of the data presented in the manuscript, as well as for the constructive suggestions to improve it.

Comments:

1. It's better to call 50,000 cells as "mini-input" rather than "low input", as there are methods that can detect chromatin interactions with fewer than 1000 cells.

We agree that in the "omics" field 50,000 cells could be considered as "mini-input". However, the "low input" term has been recently used to name other 3C-based methods (*e.g.*, Low-input Capture-C¹ or HiCuT²) that use a similar range of cells as starting material. To be more accurate, we have clearly described our method as a "mini-input" approach in the manuscript (page line numbers **68** and **353**). However, since this modification affects the name and acronym of our new method that are also in our on-going patent, we would be very grateful if you could allow us not to modify these.

2. The biological findings are mostly expected or known. To demonstrate the advantages of the new method, the authors should add more references and compare it to existing C-based methods used for detecting the 3D genome of blood lineage and for detecting cancer alterations.

We thank the reviewer for the useful suggestion to improve the manuscript. Following the advice, we have incorporated more references and performed a side-by side comparison of liCHi-C data with data of 3-C based methods of blood related samples generated using:

- Low-C³ (50K human naïve B cells)
- TagHi-C⁴ (200, 500, 1k and 1M of GM12878 human lymphoblastoid cells; 1k of mouse CLP and 1k of mouse CMP)
- Hi-C using a four-cutter enzyme⁵ (2M human naïve B cells)
- Hi-C using a six-cutter enzyme⁶ (40M human naïve B cells)
- PCHi-C⁶ (40M human naïve B cells)

Unfortunately, it was not possible to include Low-input Capture-C¹ in the study since this method generates interaction profiles rather than significant loops. Specifically, we processed Hi-C, TagHi-C and Low-C data with HiCUP⁷ as detailed in the methods section of the manuscript, using the human (GRCh38.p13) and mouse reference genomes (GRCm39). Loop calling was performed with three different methods: HICCUPS⁸ (1.22.01), Mustache⁹ (1.2.7) and HiCEXplorer¹⁰ (3.7.2). All loop callers were used with default parameters on Knight-Ruiz normalized matrices at 5 kb resolution and a maximum loop distance of 8 Mb. Specific parameters for HICCUPS were: `-cpu --ignore_sparsity`; for Mustache, the p-value threshold was set to 0.05.

Although these methods have provided fundamental insights about the role and regulation of spatio-temporal genome architecture in rare cell populations, these methods are very limited with respect to the detection of promoter interactions at high resolution (5Kb resolution) (**Extended Data Fig. 3a-b and Supplementary Information Table 2**). Indeed, this limitation is discussed in the paper that described Low-C methodology³. Collectively, these results demonstrate that liCHi-C outperforms other existing C-based methods at genome-wide and promoter-wide detection of potential gene regulatory interactions at high resolution using low-cell numbers. Therefore, liCHi-C is a worthwhile addition to the exiting 3D genome toolkits since it complements other outstanding C-based methods.

3. Capture-based C methods are currently not widely adopted. Can authors provide

information on the cost, time expenses and adoptability of the new method?

Up to 12 libraries can be generated in 6 days with a total cost of 1500 euros per library, which also include sequencing expenses. This information has been included in the manuscript (page line numbers **355-356**).

Reviewer #2 (Remarks to the Author):

In their paper, Tomas-Daza et al. describe a variant of promoter capture Hi-C (PCHi-C) suitable for low input (*i.e.*, a minimum of 50k cells) biological samples. Multiple variants of C technology suitable for low cell numbers have emerged throughout the past years (*e.g.* <https://www.nature.com/articles/s41467-018-0696103>, <https://academic.oup.com/nar/article/45/22/e184/46535401>), and this paper adds another technology to this line-up. The work is technically sound - both from wet lab and computational perspectives - and is primarily focused on benchmarking liCHi-C using a set of analyses very similar to previous work from this group using PCHi-C (Javierre et al. Cell 2016), with the addition of an analysis of topological alterations in two leukemia samples. While the manuscript is interesting the part of the community that used PCHi-C as this work shows that it can be applied to 250-500k cells, the technology presented here has several limitations that makes it difficult for liCHi-C to compete with established methods such as Low-C and low-input Capture-C (detailed below). Specific comments:

We thank the reviewer for considering our work technically sound. However, we respectfully disagree on the limitations of liCHi-C over other existing outstanding methods (Low-C³ and low-input Capture-C¹). Both methods have provided fundamental insights about the role and regulation of spatio-temporal genome architecture in rare cell populations. Nevertheless, the data generated by these outstanding methods are different from liCHi-C data due to the following reasons:

- Our new method liCHi-C can generate **high-resolution genome-wide promoter interactome maps that allow for the identification of significant regulatory loops between promoters and distal regulatory elements** (*e.g.*, enhancers). Specifically, it simultaneously interrogates the interactome of **31,253 annotated promoters in the genome at the single restriction fragment resolution** that ranges from ≈ 4096 bp, if using six-cutter restriction enzyme (*e.g.*, HindIII), to ≈ 254 bp, if using a four-cutter enzyme (*e.g.*, MboI).
- Low-C method³ allows for the generation of high-quality genome-wide chromatin conformation maps, which is fundamental to study broad topological

properties such as topologically associated domains (TADs) or compartments. However, Low-C **has limitations on the detection of *de novo* loops between gene promoters and distal regulatory elements** (e.g., enhancers) since it generates libraries with limited resolution (which ranges from 25kb to 100kb) due to their constrained complexity. Indeed, this limitation is commented in the paper describing Low-C, at the end of the discussion (please, see discussion section and "Pre Review File" included on the additional information of the paper³). Consequently, to overcome this limitation the authors propose to combine their method with a capture strategy, as liChi-C would be.

- Low-input Capture-C¹ enables the generation of high-quality promoter regulatory interaction profiles. However, this method **only allows the study of few promoters simultaneously**. Indeed, in the manuscript they demonstrate that Low-input Capture-C allows to interrogate the interactome of 5 gene promoters. Besides, Low-input Capture-C generates **high resolution interaction profiles rather than statistically significant loops**.

Consequently, liChi-C is complementary to Low-C and low-input Capture-C. Indeed, liChi-C has allowed us to provide novel mechanistic insights about the role of promoter-centric genome organization in early human hematopoiesis by identifying a dynamic enhancer repertoire controlling the transcription of each gene. This data has enabled us to identify novel genes and pathways potentially altered by non-coding variants that confer susceptibility to leukemia development and new leukemia-specific promoter-enhancer loops related and non-related with structural variants.

1. Supp.Fig.2a. shows that the number of significant interactions detected drops substantially in merged replicate samples when using <250k cells, with <50% interactions being detected in merged 50k replicates as compared to the two single replicates. This suggests that reproducibility takes a big hit when one applies PChi-C to relatively low cell numbers, which is reflected by separate clustering of 50k/100k samples from the rest in Fig.1c and Supp.Fig.2d. Reproducibility between replicates is assessed in the paper using an SCC correlation score, which appears to be rather high, but this doesn't seem to match with the numbers shown in Supp.Fig.2a. The authors

should address this issue and further explore reproducibility at the level of individual interactions in their 50k/100k samples. For example, they should show side-by-side comparisons of biological replicate interaction tracks at selected loci (*e.g.*, those shown in Supp.Fig.2e-g).

We agree with the reviewer that given the decrease in contacts in the merged samples; one should expect a decrease in reproducibility between biological replicates. In fact, we do see a small decrease in reproducibility in the 50k and 100k biological replicates (**Fig. 1b** and **Extended Data Fig. 1f**). It is also true, as **Fig. 1b** shows, that we have found a positive association between reproducibility and the number of cells. However, the slope of this correlation is very flat and the 50k replicates still show high reproducibility. However, the goal of our method and of this article is precisely to show that even if we are working with few cells, and few significant interactions, what we see is highly reproducible. Indeed, in all cases biological replicates cluster according to cell numbers and the distance between biological replicates does not correlate with the starting material replicates (**Fig. 1c**). In fact, in this PCA plot, distances between biological replicates of 50k or 100k replicates are not significantly larger or smaller than in other replicates using higher cell numbers.

As suggested by the reviewer, side-by-side comparisons of liCHi-C biological replicates 1 and 2 and the merged samples (m) interaction tracks at selected loci from Extended Data Fig.2e-g and Fig. 1d are summarized in **Extended Data Fig. 3a-b** of the revised version of the manuscript.

Most 3-C-based methodologies, excluding liCHi-C and PCHi-C, combine reads of biological replicates to identify topological properties rather than performing the analysis at the biological replicate level first, and then performing a differential analysis. Therefore, any issue regarding reproducibility between biological replicates is not as evident as in liCHi-C data. For this reason, we have benchmarked the reproducibility of liCHi-C and other low input 3C-based methods on the reproducibility of the detection of significant interactions at biological replicate level. Specifically, we have performed a side-by side comparison of liCHi-C data with data from 3-C based methods of blood related samples generated using:

- Low-C³ (50k human naïve B cells)

- TagHi-C⁴ (200, 500, 1k and 1M of GM12878 human lymphoblastoid cells; 1k of mouse CLP and 1k of mouse CMP)
- Hi-C using a four-cutter enzyme⁵ (2M human naïve B cells)
- Hi-C using a six-cutter enzyme⁶ (40M human naïve B cells)
- PCHi-C⁶ (40M human naïve B cells)

Unfortunately, it was not possible to include Low-input Capture-C¹ in the study since this method generates interaction profiles rather than significant loops. We processed Hi-C, TagHi-C and Low-C data with HiCUP⁷ as detailed in the methods section of the manuscript, using the human (GRCh38.p13) and mouse reference genomes (GRCm39). Loop calling was performed with three different methods: HICCUPS⁸ (1.22.01), Mustache⁹ (1.2.7) and HiCExplorer¹⁰ (3.7.2). All loop callers were used with default parameters on Knight-Ruiz normalized matrices at 5 kb resolution (which is similar to the resolution of liChi-C data generated with a six-cutter enzyme) and a maximum loop distance of 8 Mb. Specific parameters for HICCUPS were: -cpu --ignore_sparsity; for Mustache, the p-value threshold was set to 0.05 (**Supplementary Information Table 2**). We have compared the number of significant loops shared between the biological replicates and the corresponding merge sample generated either by liChi-C or the other existing methods. Although PCHi-C overcomes liChi-C in both, reproducibility and number of significant interactions, liChi-C has higher reproducibility (**Fig. A top**) and detects between 1 to several orders of magnitude more significant interactions (**Fig. A bottom**) than the other 3C-based methods. These results are also illustrated on **Extended Data Fig. 3a-b**

Fig. A. Number of significant interactions called at the biological replicate and the merge level (bottom) and percentage of significant interactions detected in both biological replicates and on the merged sample (top). Significant loops from liCHi-C were called by CHiCAGO at the HindIII restriction fragment resolution (≈ 4096 bp). Loops from Low-C, TagHi-C and Hi-C were called by HICCUPS (H), Mustache (Mu) and HiCExplorer (HE) at the 5kb resolution using standard parameters. Methods and cell numbers used for the library generation are indicated at the bottom of the bar plot. k (thousand) M (million).

2. In Fig.3, the authors use 500k cells for their analyses of human hematopoietic cells. While it indeed seems that liCHi-C works well on 500k cells, it's much clear (see comment above) whether it really works well on actual clinically compatible low cell numbers, *i.e.* <100k. Here, competing technology such as low-input Capture-C (<https://academic.oup.com/nar/article/45/22/e184/4653540>) appears to outcompete liCHi-C, as this method produces good quality interactions profiles on as few as 10k-20k cells. The impact of liCHi-C presented in its current form is therefore likely to be limited, unless the authors can substantiate liCHi-C's performance in the low cell spectrum.

As we previously discussed, low-input Capture-C does not outcompete liCHi-C since it only provides promoter interactome profiles of few (*e.g.*, five) gene promoters and generates high resolution interaction profiles rather than statistically significant loops. Although liCHi-C needs more cells (50k or more) as starting material than low-input Capture-C, liCHi-C simultaneously interrogates the interactome of 31,253 annotated promoters of the genome and identifies significant loops. For these reasons, both methods are complementary. Besides, we are not aware of any methodology that provides this information in a low input manner. Although not all clinically relevant cell types are currently compatible with liCHi-C (*i.e.*, more than 50k cells can be isolated), our new method significantly expands our capacity to identify genome-wide and promoter-wide, regulatory promoter loops at high resolution.

3. A drawback of the current liCHi-C protocol is that it still relies on a 6-cutter restriction enzyme (HindIII), while the entire field now used 4 bp cutter such as DpnII and MboI (or even MNase) to generate profiles with a much improved resolution. HindIII generated promoter interaction profiles are unable to pick up short-range interactions in the <20-25kb range, resulting in technology that is 'blind' to this range, which is known

to frequently harbour relevant gene-regulatory interactions. Competing technologies such as low-input Capture-C all use 4-bp cutters and do not suffer from this problem. Impact of the paper would be increased if the authors could show that liCHi-C can be adapted for use with 4 bp cutting restriction enzymes.

We thank the reviewer for the suggestion. We agree that the use of a 6-cutter restriction enzyme reduces sensitivity to detect very short-range interaction. However, the use of a 4-cutter restriction enzyme compromises the detection of extremely long-range interactions, which also play a role in gene regulatory functions. In this study we used a 6-cutter restriction enzyme because we aimed to compare these against the existing PCHi-C libraries that were also generated with the same 6-cutter restriction enzyme. However, liCHi-C can be coupled with both a 4-cutter, and a 6-cutter restriction enzyme.

To demonstrate this versatility, we have isolated blasts from a B-ALL patient, and we have generated two high resolution liCHi-C libraries using a 4-cutter restriction enzyme (MboI) and a 6-cutter cutter restriction enzyme (HindIII) (**Fig. 6, Extended Data Fig. 8h-k** and **Supplementary Information Table 1**). liCHi-C libraries generated with MboI detected 1.78 times more of significant interactions (**Extended Data Fig. 8j**), which were characterized by having half of the median linear distance between promoters and their interacting regions (**Fig. 6c**). Indeed, although the shortest significant interaction was similar in both libraries (2574bp for HindIII and 1939bp for MboI), the highest frequency of interactions was found at a distance 2.15 times larger for HindIII restriction (**Fig. 6c**). This data, illustrated by the promoter interactome of the *DDX41* (**Fig. 6d**), a DEAD box RNA helicase associated with B-ALL and other blood malignancies¹¹, demonstrates that the use of a four-cutter restriction enzyme increases the power to detect short-range interactions and compromises the detection of the long-range ones. Taken together, our results demonstrate liCHi-C capability to provide fundamental and clinical insights about gene-regulatory interactions at different levels of resolution.

Reviewer #3 (Remarks to the Author):

Tomas-Daza et al outline the development and potential utility of their improved, low input promoter capture HiC (PCHiC, Javierre et al., Cell, 2016) method liCHiC. They reveal the promoter centric 3D genome architecture of the haematopoietic lineage, overlay 3D architecture with gene expression, epigenetic marks and disease associated genetic variants all with their low input capture method.

I have a few concerns.

We thank the reviewer for the constructive feedback.

Major concerns.

1. Capture HiC libraries historically have very high duplication percentages. What are the duplication rates in the samples from this work? Table 1 suggests they are perhaps as high as 90% in the 50K B cell samples but these numbers are not as apparent as the other emboldened library statistics. As duplication has a major impact on the usability and cost of these libraries, I feel that the duplication percentages should be both outlined in the relevant section of text (line 100-103) and clearly stated in Table 1.

We have updated **Supplementary Information Table 1** with this information (**row 43**), which is also outlined in the relevant section of text (line **103-105**). Just to note, the 50k and the 40M samples have a median of 86% and 12% duplication rates, respectively.

Fig. B summarizes all this data.

Fig. B. Percentage of duplication rates of liCHi-C libraries. liCHi-C libraries were generated using 50,000 (50k), 100,000 (100k), 250,000 (250k), 500,000 (500k) and 1 million (1M) of human naive B cells. PCHi-C libraries were generated using 40M of human naive B cells.

2. It is also not clear from Table 1 what the final number of usable reads are per sample. This would be useful to have clearly outlined to enable a reader to determine, for example, how many reads reveal ~100,000 interactions.

We apologize for the lack of clarity. This information has been also included in **Supplementary Information Table 1 (row 52)**. For instance, 42,561,081 and 39,089,588 captured unique valid reads per biological replicate generated 94,664 and 100,708 significant promoter interactions respectively. **Fig. C** summarizes all this data.

Fig. C. Number of unique valid captured valid reads (left) used to call significant promoter interactions (right) at liCHi-C libraries. liCHi-C libraries were generated using 50,000 (50k), 100,000 (100k), 250,000 (250k), 500,000 (500k) and 1 million (1M) of human naive B cells. PCHi-C libraries were generated using 40M of human naive B cells.

3. The number of cells used per library for all samples is also not immediately apparent, either within the text or Table 1. For example, how many HSCs or CLPs were used for generate each library? Was it 500K? How was this decision made? How many human fetal livers were required to obtain these? Given the paper hinges on the amount of input required for the technology, not providing, or making this this information difficult to find is a significant oversight.

We apologize since we did not realize that this information was difficult to find. Excluding the naïve B cell titration and the new B-ALL libraries from patient 3 incorporated in the new version of the manuscript, the rest of liCHi-C libraries were generated using 500,000 (500k) cells. 250,000 (250k) blasts from B-ALL patient 3 were used for the preparation of each liCHi-C libraries generated with HindIII or MboI to benchmark liCHi-C resolution. This information has been made more apparent in the

manuscript (row 2 Supplementary Information Table 1 and lines 153, 268 and 320 of the revised manuscript).

The decision regarding the cell number per biological replicates was made as a compromise between three criteria: i) cell number availability and estimation of abundance of clinically relevant cell types; ii) price of cell isolation and liCHi-C library generation and sequencing; and iii) estimation of the number of detected significant promoter interactions.

HSCs, CLPs and CMPs were simultaneously obtained from 4 donations of 15- to 22-week-old human fetal liver and fetal bone marrow (2 donations per biological replicate). This information has been included in the method section (line 605).

4. While I am aware the comparative analysis was performed on merged samples, I would like to see the replicates shown in Fig 2B.

Supplementary Data Fig. 4f of the new version of the manuscript (also included on the left side of **Fig. D**) summarizes a PCA analysis at the biological replicate level of liCHi-C (500k) and PCHi-C (40M) data obtained from the same cell types. We have also incorporated in the analysis the merged liCHi-C and PCHi-C libraries (nB, nCD8, nCD4, Ery, Mon and MK) and the liCHi-C libraries for the remaining cell types (HSCs, CLPs and CMPs) at the biological replicate and merged levels (right part of **Fig. D**).

Fig D. Clustering of liCHi-C and PCHi-C biological replicates **Left:** Principal component analysis (PCA) of CHiCAGO significant interactions called in each biological replicate of liCHi-C (o) and PCHi-C (*). **Right:** PCA of CHiCAGO significant interactions called in each biological replicate and merged sample of liCHi-C (o) and PCHi-C (*). 500,000 and 40 million cells were used for liCHi-C and PCHi-C, respectively. Hematopoietic stem cell (HSC), common myeloid progenitor (CMP), common B cell lymphoid progenitor (CLP), megakaryocytes (MK), monocytes (Mon), erythroblast (Ery), naïve B cell (nB), naïve CD4⁺ cells (nCD4) and naïve CD8⁺ cells (nCD8).

5. While later figures show the genome-wide (if epigenetic state specific) relationship between expression and interactivity, I would like to see the impact of the differences in interactivity shown in Fig 2E-G on the expression of the specific genes shown. A simple overlay with RNA-Seq would suffice.

Fig. E. Interplay between gene expression and interactivity. Distribution of RPKM values for genes in which their promoter connectivity is represented on Fig. 2 E-G (A) and Supplementary Fig 5E-F (B) compared to expression of a publicly available set of housekeeping genes from¹². Promoter-centered interaction landscape are included at the bottom of each panel.

6. In the Blockshifter analysis of GWAS SNPs and PIRs Line 208 states the PIRs analyses are cell-type specific. How specific is this? For example, it was shown that CD4 and CD8+ T cells have largely overlapping 3D architectural profiles, thus should have less cell-type specific PIRs than say monocytes which possess quite unique PIR profiles. Does this mean that the analysis is underpowered to detect T cell-dependent disease profiles? Clarification would be valuable.

We apologize for not expressing ourselves with total accuracy. The use of cell-type specific is no accurate. This has been corrected in the manuscript (lines **223-225**). We mean that PIRs defined in a given cell type (independently of whether these are shared or not across cell types) are enriched for genetic variants associated with traits or diseases relevant to the given cell type. This means that, for instance, if we compute for enrichment of non-coding SNPs regulating red blood cells, these SNPs are more enriched in PIRs defined in erythroblasts than PIRs of lymphocytes. Therefore, the analysis does not underpower the detection of any trait or disease.

Minor concerns

1. I think many of the conclusionary statements within the paper are overstated. I include a few examples below.

We apologize for the overstated statements and the minor grammatical errors. These have been corrected in the manuscript. The corrections and the modified lines on the manuscript are also specified below.

2. Line 145 “independently of the method used or the starting number of cells (Extended Data Fig. 3f).” 1) I do not believe that the data show independence from starting cell number as all all samples are either 500K (I believe, see above) or 40M. A reader could easily confuse this with the previous experiments titrating cell number. More accurate language should be used. 2). Also, the figure panels are referred to out of order.

Line 159-160: Moreover, promoter interactomes clearly separated cell types independently of the method used (Extended Data Fig. 2f).

3. Line 150 “demonstrated that promoter interactomes were highly reproducible and dynamic (Fig. 2b).” – data does not show dynamics at all, I would suggest changing “dynamic” to “cell-type specific”

Line 164-165: A PCA of interaction scores demonstrated that promoter interactomes were highly reproducible and cell-type specific.

4. Line 186: “regulatory elements for each gene in rare cell types hitherto undeterminable, and suggest that” – ‘hitherto undeterminable’ is overstated. These interactions may have been detected previously in HiC or other methods.

Line 197-203: Collectively, these results, exemplified by the transcriptional regulation of the T cell-specific gene GATA3¹³ (Fig. 3b), the B cell-specific gene PAX5¹⁴ (Extended Data Fig. 4e), and the HSC-specific gene CD34¹⁵ (Extended Data Fig. 4f), demonstrate the capacity of liChi-C to identify distal regulatory elements for each gene in rare cell types, and suggest that promoter-associated regions are enriched in distal regulatory elements that mirror the cell-type specificity of the interacting gene’s expression.

5. Line 197: “these results illustrate the power of liChi-C to expose new aspects on the diversity of factors and mechanisms regulating genome architecture in physiological and pathogenic settings” – the authors overlay a number of histone marks with regions of interactivity. This does not reveal factors or mechanism regulating 3D genome architecture, and certainly not in disease. It is at most an association, albeit a logical and strong one. Please reword.

Line 211-213: Collectively, these results illustrate the capacity of liCHi-C to suggest, after further functional validation, new aspects on the diversity of factors and mechanisms regulating genome architecture.

6. Line 244: “Most of the ligation events detected by proximity-ligation methods, such as liCHi-C, occur between contiguous sequences in the linear genome” – This is untrue. While it is true that the majority of ligation events occur between regions that are less than 10kb apart in the linear genome, very few relink the genome contiguously. Please reword.

Line 260-263: Most of the ligation events detected by proximity-ligation methods, such as liCHi-C, occur between sequences in close proximity in the linear genome and the frequency of these events decreases logarithmically with genomic distance that separates them.

7. Line 334: “liCHi-C enables the study of primary samples, thus addressing a key limitation to apply this type of analysis to clinical samples”. Please clarify that this refers to the comparison to PCHiC specifically. HiC is readily performed on primary human cells.

Line 375-376: liCHi-C enables the study of primary rare samples, thus addressing a key limitation of PCHi-C to apply this type of analysis to clinical samples.

8. There are also a handful of minor grammatical errors within the manuscript.

References

1. Oudelaar, A. M., Davies, J. O. J., Downes, D. J., Higgs, D. R. & Hughes, J. R. Robust detection of chromosomal interactions from small numbers of cells using low-input Capture-C. *Nucleic Acids Res.* **45**, e184–e184 (2017).
2. Sati, S. *et al.* HiCuT: An efficient and low input method to identify protein-

- directed chromatin interactions. *PLoS Genet.* **18**, e1010121–e1010121 (2022).
3. Díaz, N. *et al.* Chromatin conformation analysis of primary patient tissue using a low input Hi-C method. *Nat. Commun.* **9**, 4938 (2018).
 4. Zhang, C. *et al.* tagHi-C Reveals 3D Chromatin Architecture Dynamics during Mouse Hematopoiesis. *Cell Rep.* **32**, (2020).
 5. Vilarrasa-Blasi, R. *et al.* Dynamics of genome architecture and chromatin function during human B cell differentiation and neoplastic transformation. *Nat. Commun.* **12**, 651 (2021).
 6. Javierre, B. M. *et al.* Lineage-Specific Genome Architecture Links Enhancers and Non-coding Disease Variants to Target Gene Promoters. *Cell* **167**, 1369–1384.e19 (2016).
 7. Wingett, S. W. *et al.* HiCUP: pipeline for mapping and processing Hi-C data [version 1; peer review: 2 approved, 1 approved with reservations]. *F1000Research* **4**, (2015).
 8. Durand, N. C. *et al.* Juicer Provides a One-Click System for Analyzing Loop-Resolution Hi-C Experiments. *Cell Syst.* **3**, 95–98 (2016).
 9. Roayaei Ardakany, A., Gezer, H. T., Lonardi, S. & Ay, F. Mustache: multi-scale detection of chromatin loops from Hi-C and Micro-C maps using scale-space representation. *Genome Biol.* **21**, 256 (2020).
 10. Wolff, J., Backofen, R. & Grünig, B. Loop detection using Hi-C data with HiCExplorer. *Gigascience* **11**, giac061 (2022).
 11. Shin, W. Y. *et al.* A novel bi-allelic DDX41 mutations in B-cell lymphoblastic leukemia: case report. *BMC Med. Genomics* **15**, 46 (2022).
 12. Eisenberg, E. & Levanon, E. Y. Human housekeeping genes, revisited. *Trends Genet.* **29**, 569–574 (2013).
 13. Ho, I.-C., Tai, T.-S. & Pai, S.-Y. GATA3 and the T-cell lineage: essential functions before and after T-helper-2-cell differentiation. *Nat. Rev. Immunol.* **9**, 125–135 (2009).
 14. Cobaleda, C., Schebesta, A., Delogu, A. & Busslinger, M. Pax5: the guardian of B cell identity and function. *Nat. Immunol.* **8**, 463–470 (2007).
 15. Hughes, M. R. *et al.* A sticky wicket: Defining molecular functions for CD34 in hematopoietic cells. *Exp. Hematol.* **86**, 1–14 (2020).

REVIEWERS' COMMENTS

Reviewer #1 (Remarks to the Author):

I do not have further comments and support the publication of this work.

Reviewer #2 (Remarks to the Author):

I appreciate the efforts made by the authors, in particular the addition of 4-bp cutter data has strengthened the manuscript. I do have one remaining issue:

- I still find the reproducibility of the 50k and 100k samples problematic, as is shown in Supp. Fig. 1a: de authors call ~twice the number of interactions in individual replicates as observed in the merged sample. This to me can only be explained by relatively poor reproducibility when using the technique with 100k or fewer cells, which is also clearly visible from the loop calls shown in Supp.Fig.3 (11.3% loops shared between replicates). I find the high SCC scores very counterintuitive (what do they mean anyways?). The fact that the 50k replicates cluster in a PCA together with sample analysing large cell numbers seems mostly driven by the fact that a much lower number of interactions is called in the 50/100k samples. I don't think this is necessarily a bad thing, it is just something I feel that the authors should be more open and honest about.

Reviewer #3 (Remarks to the Author):

My major concerns have been well addressed. I feel the manuscript is now a more accessible and clear work. Thank you

Second round of point-by-point responses to Reviewers' comments (NCOMMS-22-19740-T).

First, we would like to thank the reviewers for the constructive criticism and suggestions to improve the manuscript.

To help with the structuring of this document, we have marked the reviewer's comments with **green typography**, while our replies are shown in normal black font. Besides, the main editions to the manuscript are also marked in **green typography** and we have included the line numbers to help the identification of these in the manuscript.

Reviewer #1 (Remarks to the Author):

I do not have further comments and support the publication of this work.

Thanks. We really appreciate the previous feedback that has significantly improved our manuscript.

Reviewer #2 (Remarks to the Author):

I appreciate the efforts made by the authors, in particular the addition of 4-bp cutter data has strengthened the manuscript. I do have one remaining issue:

- I still find the reproducibility of the 50k and 100k samples problematic, as is shown in Supp. Fig. 1a: the authors call ~twice the number of interactions in individual replicates as observed in the merged sample. This to me can only be explained by relatively poor reproducibility when using the technique with 100k or fewer cells, which is also clearly visible from the loop calls shown in Supp.Fig.3 (11.3% loops shared between replicates). I find the high SCC scores very counterintuitive (what do they mean anyways?). The fact that the 50k replicates cluster in a PCA together with sample analyzing large cell numbers seems mostly driven by the fact that a much lower number of interactions is called in the 50/100k samples. I don't think this is necessarily a bad thing, it is just something I feel that the authors should be more open and honest about.

We thank the reviewer for the comment. We agree with the reviewer that reproducibility for 50k and 100k samples are lower than libraries using higher cell numbers as starting material (**Fig. 1b** and **Extended Data Fig. 1f**). SCC analysis also support it. However, it is important to keep in mind the differences in resolution here that can explain the misinterpretation. CHICAGO peak works at the restriction fragment resolution, while SCC works at the level of 100kb bins. To avoid misunderstandings, this information is included on the manuscript (**Lines 107, 895 and 1056**). As shown in the example, **Fig. 1d**, while many single interactions are lost when lowering the number of cells, the general interaction pattern is maintained. With a lower number of input cells, the probability that the exact pair of restriction fragments is found significantly enriched to represent a given chromatin interaction is necessarily much lower (and with a quadratic factor). This is the reason why, with a low number of input cells, we find high SCCs at 100kb while we lose a significant amount of consensus interactions between replicates.

To confirm this hypothesis, we have plotted how the overlap between replicates varies when reducing the resolution. In the plot below we can see that with low number of cells 50K or 100K, the increase in overlap coefficient is dramatic when lowering the resolution, while with higher number of cells the effect of changing the resolution is close to null (even reversed with >1M cells).

Overlap-coefficient of CHICAGO interaction (score > 1) between replicates. Measure is done on replicate pairs with different input sizes (in the X axis). The overlap is defined using different resolutions (represented by the dotted lines in gray scales): from exact match (1b; darker line), to distant match (up to 100kb allowed between interaction anchors; lighter gray line). Each observation is represented by a Venn-Diagram to show the number of interactions and the intersection of the replicates.

Reviewer #3 (Remarks to the Author):

My major concerns have been well addressed. I feel the manuscript is now a more accessible and clear work. Thank you

We want to thank you for the constructive suggestions.